## TOOLS AND RESOURCES

# The desmoglein 2 interactome in primary neonatal cardiomyocytes

Yang Li[1], Alexandra P. Campbell[1,*], Sahana Balasubramanian[1,*], Xuemei Zeng[2], Emma Porter[1], Pamela S. Cantrell[2], Mai Sun[2], Alexa L. Mattheyses[3] and Adam V. Kwiatkowski[1,‡]

## ABSTRACT

Mechanical coupling and chemical communication between cardiomyocytes are facilitated through a specialized adhesive structure called the intercalated disc (ICD). The ICD is essential for heart organization and contraction. Yet, the network of adhesion, adaptor and signaling proteins that form the ICD remains poorly defined. Here, we combined proximity labeling and quantitative mass spectrometry to identify proteins associated with the desmosomal cadherin desmoglein 2 (DSG2), in cultured neonatal cardiomyocytes. We identified over 300 proteins in the DSG2 interactome, half of which are shared with the N-cadherin (CDH2) interactome in cardiomyocytes. Proteins unique to DSG2 include connexin 43 and the plakin family of cytolinker proteins. Comparison of the cardiomyocyte DSG2 interactome with the interactomes of desmosomal proteins from epithelia revealed few shared proteins. In cardiomyocytes, plakoglobin and plakophilin 2 (PKP2) were the most abundant shared proteins between the DSG2 and CDH2 interactomes. We show that PKP2 is a dynamic protein whose membrane recruitment in cardiomyocytes is dependent on tension. Our analysis of the DSG2 interactome provides a crucial new dimension to the proteomic atlas of the essential molecular complexes required for cardiomyocyte adhesion.

KEY WORDS: Desmosome interactome, Cardiomyocyte, Plakophilin 2, Cell–cell adhesion, Desmoglein 2, Intercalated disc

## INTRODUCTION

The heart is formed from a massive, interconnected network of individual cardiac muscle cells, known as cardiomyocytes, that function collectively to pump blood throughout the body. Cardiomyocytes are connected end-to-end by a specialized adhesive structure called the intercalated disc (ICD), which enables both mechanical and electrochemical coupling between cardiomyocytes (Nielsen et al., 2023; Pruna and Ehler, 2020). The ICD comprises three adhesion complexes: adherens junctions (AJs) and desmosomes, which link the actin and intermediate filament (IF) cytoskeletons

of adjoining cells, respectively, and gap junctions, which are membrane pores that permit the exchange of ions and small metabolites between cells. The establishment, maintenance and function of all three adhesion complexes are essential for proper heart function, as mutations in the proteins that form these complexes are linked to heart disease, notably arrhythmogenic cardiomyopathy (AC) (Nielsen et al., 2023). Yet, we lack a complete understanding of the proteins that form these adhesion complexes and how they work together to establish and maintain intercellular adhesion in the heart under demanding mechanical conditions unique to this organ.

The AJ and desmosome are well-studied adhesion complexes, best understood for their function in epithelia (Campas et al., 2024; Rubsam et al., 2018; Yeruva and Waschke, 2023). The core of the cardiac desmosome is formed by a pair of transmembrane proteins, desmosomal cadherins, desmoglein 2 (DSG2) and desmocollin 2 (DSC2, Fig. 1A). The extracellular (EC) domains of DSG2 and DSC2 engage in homotypic and heterotypic interactions with the EC domains of desmosomal cadherins on opposing cells. Heterotypic binding (i.e. DSG2 on one cell binds DSC2 on the adjacent cell) is postulated as the primary adhesion mechanism. Cytoplasmic linker proteins mediate the linkage of desmosomal cadherins to IFs (Harmon and Green, 2013). Plakoglobin (JUP) and plakophilin 2 (PKP2) are armadillo (ARM) repeat domain-containing proteins that bind directly to the tails of DSG2 and DSC2. JUP and PKP2, in turn, recruit desmoplakin (DSP), which links the desmosome complex to IFs (Fig. 1A).

AC is often described as a desmosomal disease, as pathogenetic variants in all desmosomal proteins are linked to AC in humans (Delmar and McKenna, 2010). Most AC-linked mutations are found in PKP2 and DSG2 (Zhang et al., 2023), underscoring the importance of both proteins in cardiovascular health. Loss of DSG2 function in the developing mouse heart prevents desmosome formation along the ICD and leads to AC (Kant et al., 2015). Likewise, deletion of extracellular (EC) regions 1 and 2 in DSG2 causes AC-like phenotypes in mice (Kant et al., 2012; Krusche et al., 2011), as does overexpression of the Dsg2 mutant N271S (corresponding to the AC-linked Dsg2 N266S mutation in humans) (Pilichou et al., 2009). Thus, DSG2 is essential for desmosome formation in cardiomyocytes, and the loss or disruption of DSG2 function leads to the development of cardiovascular disease.

In polarized epithelia, the desmosome organizes as a distinct, well-defined structure below (basal to) the AJ and connects the keratin networks of adjacent cells (Broussard et al., 2020; Nekrasova and Green, 2013; Yeruva and Waschke, 2023). Cardiomyocytes, in contrast, lack apical-basal polarity, and AJs and desmosomes are interspersed along the ICD. AJ and desmosome components can also be intermingled at cardiomyocyte cell–cell contacts, especially during ICD development (Borrmann et al., 2006; Franke et al., 2006, 2007). Furthermore, there is evidence that the desmosomal

[1]Department of Cell Biology, University of Pittsburgh School of Medicine, Pittsburgh, PA 15261, USA. [2]Biomedical Mass Spectrometry Center, University of Pittsburgh Schools of the Health Sciences, Pittsburgh, PA 15261, USA. [3]Department of Cell, Developmental, and Integrative Biology, The University of Alabama at Birmingham, Birmingham, AL 35294, USA.
*These authors contributed equally to this work

‡Author for correspondence (adamkwi@pitt.edu)

A.L.M., 0000-0002-5119-7750; A.V.K., 0000-0003-3997-7057

*Journal of Cell Science*

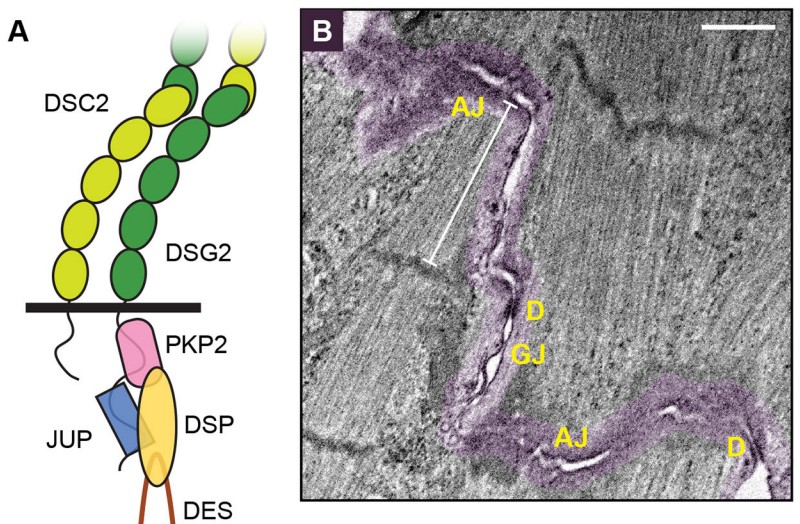

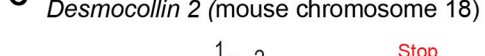

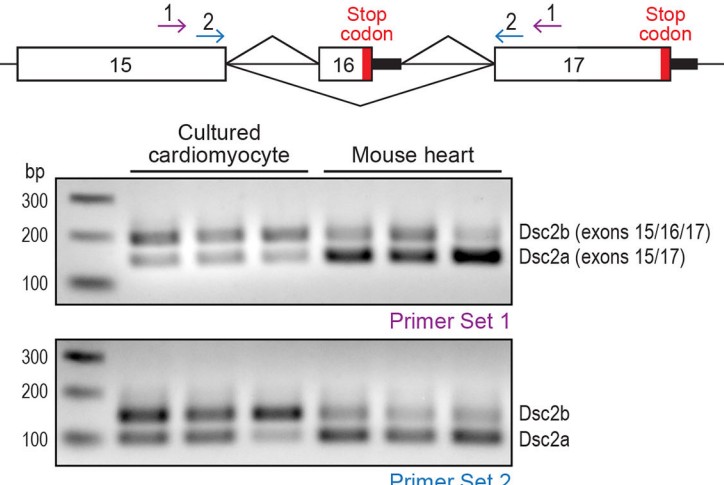

**Fig. 1. Desmosome organization at neonatal cardiomyocyte cell–cell contacts.** (A) Cartoon schematic of the cardiac desmosome showing desmocollin 2 (DSC2), desmoglein 2 (DSG2), plakophilin 2 (PKP2), plakoglobin (JUP), desmoplakin (DSP) and desmin (DES). (B) Thin section electron microscopy image of neonatal cardiomyocyte cell–cell junction. The cell–cell contact is highlighted in purple. Desmosomes (D), adherens junctions (AJ) and a gap junction (GJ) are labeled. White line marks a terminal sarcomere. Image representative of at least four experimental repeats. Scale bar: 1 μm. (C) Top, cartoon schematic of exon organization and splicing at the 3′ end of the *Dsc2* gene on mouse chromosome 18. Splicing of exons 15 and 17 produces variant Dsc2a with a longer C-terminal cytoplasmic tail. In contrast, splicing of exons 15, 16 and 17 introduces a new stop codon (red bar) to create variant Dsc2b with a truncated C-terminal tail. PCR primer pair locations for set 1 and set 2 are shown as arrows. Bottom, PCR results from cDNA isolated from cultured neonatal cardiomyocytes (three independent preps) and mouse hearts (three individual hearts).

protein PKP2 binds a core component of the cardiac AJ, αT-catenin (CTNNA3), to form a unique hybrid junction at the ICD termed the area composita (Goossens et al., 2007; Li et al., 2012; van Hengel et al., 2013). Previously, we have used proximity proteomics to define the N-cadherin (CDH2) interactome in neonatal cardiomyocytes and identified adaptor and adhesion proteins unique to the cardiomyocyte AJ (Li et al., 2019). Among the most abundant hits were two proteins associated with the desmosome, JUP and PKP2. The enrichment of desmosomal proteins at the AJ could reflect the proximity of the two adhesion complexes at developing junctions or the mixing of components in hybrid junctions. The blending of AJ and desmosomal proteins, along with the existence of a hybrid junction, underscores the uniqueness of cardiomyocyte cell–cell adhesion and the potential for mechanical and signaling crosstalk between desmosomes and AJs.

Here, we define the DSG2 interactome in cardiomyocytes, gaining insight into the composition and organization of cardiomyocyte adhesion complexes. We demonstrate that cardiomyocyte desmosome proteins are largely stable, albeit with varying dynamics that might reflect distinct roles at junctions. We used proximity proteomics to identify proteins associated with DSG2. Approximately 50% of the DSG2 interactome overlapped with the CDH2 interactome (Li et al., 2019), reflecting the proximity of the two complexes and shared

cellular pathways. Nonetheless, we identified 153 proteins unique to DSG2, suggesting that the two adhesion proteins assemble distinct complexes at developing contacts. Finally, we provide evidence that PKP2 is a primary component of both the desmosome and the AJ in cardiomyocytes. Our results provide insight into the cardiomyocyte desmosome and the molecular complexes assembled at cardiomyocyte cell–cell contacts.

## RESULTS

### The desmosome in neonatal cardiomyocytes

Neonatal cardiomyocytes can reestablish cell–cell junctions when grown in culture (Li et al., 2019; Merkel et al., 2019). Newly formed junctions contain the three ICD adhesion complexes – AJs, desmosomes and gap junctions – intermingled along the contact membrane (Fig. 1B). The AJ is the primary complex in these nascent junctions (or pre-ICD), where it functions to couple the cortical cytoskeletons and link contractile myofibrils of adjacent cardiomyocytes. Desmosomes are organized adjacent to and between myofibril-coupled AJs along the pre-ICD and are often proximal to small gap junctions (Fig. 1B). Although neonatal cardiomyocytes lack mature ICDs and the classic bipolar, rod-shaped morphology observed in the adult heart, the pre-ICD shares many of the organizational and architectural hallmarks of the ICD.

Two isoforms of DSC2 are generated by alternative splicing (Fig. 1C) (Parker et al., 1991). Isoform DSC2a is recognized as the canonical protein, a 901-amino-acid (aa) protein formed by splicing exons 15 and 17. Isoform DSC2b is formed by splicing exons 15 and 16, which removes a large part of the C-terminal tail (aa 837–901), including part of the catenin-binding domain (CBD), and replaces it with a short, unique 11 aa peptide (aa 837–847) (Parker et al., 1991; Troyanovsky et al., 1994). To determine which Dsc2 isoforms are expressed in the heart, we isolated RNA from cultured neonatal cardiomyocytes and 3-week-old mouse hearts. We used two-step RT-PCR to amplify the *Dsc2* cDNA region flanking exon 16 (Fig. 1C). If exon 16 were spliced downstream of exon 15 (as in Dsc2b), a fragment of ~190 bps (primer set 1) or 140 bps (primer set 2) would be observed. However, if exon 16 were absent (as in Dsc2a), a fragment of ~150 base pairs (bps, primer set 1) or ~100 bps (primer set 2) would be generated. We observed both Dsc2a and Dsc2b isoforms in all cDNA samples (Fig. 1C). However, Dsc2b was slightly enriched in cardiomyocyte samples, whereas Dsc2a was the predominant form in heart samples (Fig. 1C).

### Desmosome protein dynamics

We next examined the dynamics of DSG2, DSC2a, DSC2b, PKP2 and DSP in cardiomyocytes. The dynamics of JUP, which binds directly to classical cadherins as part of the AJ, were published previously by our group (Li et al., 2019). Fluorescent protein (FP)-tagged proteins were transfected individually into cultured neonatal cardiomyocytes. As expected, all fusion constructs localized to cell–cell contacts (Fig. 2A). Fusion protein dynamics were measured by fluorescence recovery after photobleaching (FRAP) in cells plated at semi-confluency and cultured for 48–72 h (Fig. 2A). Fluorescence recovery over 10 min was quantified, plotted and a double exponential curve fit to the data (Fig. 2B). The mobile fraction and recovery halftimes of the fast and slow pools were calculated from the fit (Fig. 2C). For all desmosome proteins, the fast pool was a relatively small portion of the mobile fraction (3–15%); thus, the slow pool represents the dynamics of most of the junction population.

We then plotted the 95% confidence interval range of the mean for each protein recovery dataset and overlaid the curve fit (Fig. 2D). We also included JUP data from our previous study (Li et al., 2019). Comparing the recovery profiles revealed that DSG2, JUP and DSP had highly overlapping curve fits and confidence intervals (Fig. 2D). Consistent with this, the calculated mobile fractions of DSG2 and DSP (26.9 and 28.8%; Fig. 2C) and JUP (26.5%; Li et al., 2019) were similar. Likewise, the calculated recovery halftimes of DSG2 (231.0 s) and JUP (237.6 s; Li et al., 2019) were nearly identical (Fig. 2C). The recovery halftime of DSP was slightly slower (288.3 s). These slow rates reflect strong associations between core components of the desmosome (Al-Jassar et al., 2013; Choi et al., 2009; Kami et al., 2009). Based on the recovery data, we conclude that DSG2, JUP and DSP form a strong, stable desmosome core in cardiomyocytes.

The dynamics of DSC2a and DSC2b differed from that of the DSG2–JUP–DSP core (Fig. 2D). The DSC2a mobile fraction (33%) was similar to that of the core proteins, indicating that most DSC2a is integrated into immobile, stable adhesions. However, the recovery halftime was slightly faster (189.5 s), suggesting that the mobile pool was more dynamic than the DSG2–JUP–DSP core (231.0–288.3 s). In contrast, nearly half of the DSC2b pool was mobile (46.3%), but this mobile fraction recovered slowly (256.5 s), similar to what was seen for the DSG2–JUP–DSP core. Differing dynamics between the DSC2 isoforms could reflect inherent differences in IF binding and/or associations with other desmosomal proteins.

PKP2 dynamics differed from all other desmosomal proteins (Fig. 2D). The large mobile fraction (53.3%) and relatively fast recovery halftime (213.5 s) indicate that PKP2 is more dynamic than the other desmosome proteins. The PKP2 mobile fraction in MDCK epithelial cells was also significantly higher than that of other desmosomal proteins (75% versus 30–50%) (Fülle et al., 2021). Distinct dynamics could reflect PKP2 functions independent of cytoskeletal coupling (Bass-Zubek et al., 2009).

Finally, we compared the desmosome protein recovery profiles to the core AJ proteins N-cadherin (CDH2), β-catenin (CTNNB1) and αE-catenin (CTNNA1) (Fig. 2E). AJ core proteins, like desmosome core proteins, had similar and overlapping recovery profiles (Fig. 2E, AJ protein recovery profiles in blue). The AJ recovery profiles fell between those of the desmosome core (DSG2–JUP–DSP) and DSC2a (Fig. 2E). This difference is reflected in AJ proteins having slightly larger mobile fractions (32.3–36.4%) and slower recovery rates (255.2–360 s) (Li et al., 2019). Notably, the FRAP results indicate that, like the AJ, the neonatal cardiomyocyte desmosome in general, and DSG2 in particular, is stable and amenable to proximity labeling.

### DSG2–BioID2 labels proteins at cardiomyocyte cell–cell contacts

We sought to define the proteins associated with the desmosome using proximity proteomics, as we did for the cardiomyocyte N-cadherin interactome (Li et al., 2019). We decided to use DSG2 as the probe because (1) exogenous DSG2 tagged at the C-terminus with EGFP was stably expressed and localized to cell–cell contacts (Fig. 2A), and (2) DSG2 dynamics were more favorable to biotin labeling (decreased mobile fraction, slower recovery halftime) than DSC2a or DSC2b (Fig. 2C). We fused the biotin ligase BioID2 (Kim et al., 2016) to the C-terminal tail of DSG2 (Fig. 3A) and cloned the DSG2–BioID2 fusion into an adenoviral expression system to make DSG2–BioID2 adenovirus. We infected primary cardiomyocytes with DSG2–BioID2 adenovirus and expressed low levels of DSG2–BioID2 for imaging and protein analysis (Fig. 3B,C). When expressed in cardiomyocytes and supplied with biotin in the medium, DSG2–BioID2 biotinylated proteins at cell–cell contacts [Streptavidin (SA) stain in Fig. 3B, asterisks mark expressing cells]. To further assess protein labeling, we precipitated biotinylated proteins from lysates of cardiomyocytes infected with either Dsg2–BioID2 or Cdh2–BioID2 and cultured with biotin. Precipitated material was separated by SDS-PAGE, blotted and analyzed for BioID2 fusion expression and biotinylation. DSG2–BioID2 was stably expressed (Fig. 3C, HA blot) and biotinylated a large pool of proteins (Fig. 3C, SA blot), similar to CDH2–BioID2.

### Quantitative proximity proteomics reveals the cardiomyocyte DSG2 interactome

Next, we used quantitative mass spectrometry (MS) to define the DSG2 interactome. For each replicate, $4{\times}10^6$ cells were infected with Dsg2–BioID2 adenovirus 24 h post plating. The next day (48 h post plating), 50 µM biotin (final concentration) was added to the medium, and the cells were harvested the following day (72 h post plating). Uninfected control samples were treated identically to Dsg2–BioID samples (i.e. 50 µM biotin was added 48 h post plating, and cells were harvested 24 h after biotin addition). In total, six Dsg2–BioID2 replicates (each containing $4{\times}10^6$ cells, for a total of $24{\times}10^6$ cells) and six control replicates (each containing $4{\times}10^6$ cells, for a total of $24{\times}10^6$ cells) were collected and analyzed.

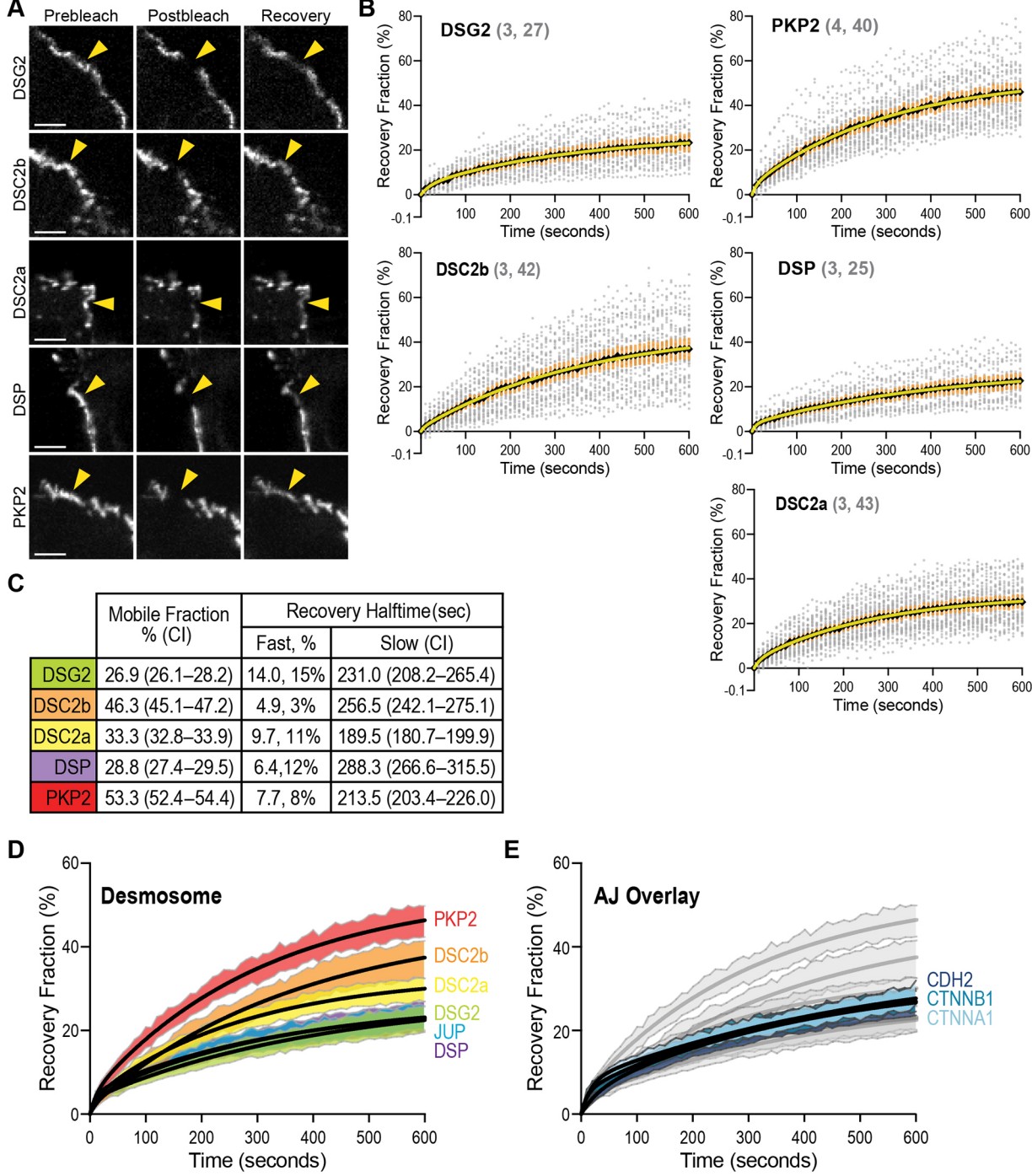

**Fig. 2. Desmosome protein dynamics.** (A) Images from FRAP experiments showing pre-bleach, post-bleach and recovery after 10 min in cardiomyocytes expressing GFP-tagged DSG2, DSC2b, DSC2a, DSP and PKP2. Yellow arrowheads point at the FRAP region along a cell–cell contact. Scale bar: 5 µm. (B) Plots of FRAP recovery fraction over 10 min. Values in gray (#, ##) indicate the number of biological replicates and FRAP contacts quantified for each protein. All data points are shown as gray circles. Black diamonds define the mean and orange vertical lines mark the 95% confidence interval at each time point. A double exponential curve was fit to the data (yellow line). (C) The mobile fraction (percentage) and recovery halftimes (seconds, sec) for the fast and slow pools. The 95% confidence interval (CI) is listed for the mobile fraction percentage and slow pool recovery rate. The fast pool percentage of the total mobile fraction is also listed. (D) Comparison of desmosome protein FRAP recovery curve fits (black lines) and 95% CIs (shaded regions). JUP data from Li et al. (2019). (E) Desmosome FRAP profiles from (D, in gray) overlaid with CDH2, CTNNB1 and CTNNA1 FRAP recovery curve fits (black lines) and 95% c.i. (blue shaded regions). AJ protein FRAP data from Li et al. (2019).

MS sample analysis revealed 5809 peptides from 924 proteins (Fig. 4A,B). When proteins with single unique peptides were excluded, the list was reduced to 687 proteins (Fig. 4B). To define Dsg2–BioID2-enriched proteins, we established thresholds of fold change ≥10 and $P<0.001$ (Fig. 4A, dashed lines). These thresholds culled the list to a final 331 proteins from 325 genes (Fig. 4B; Table S1).

The relative abundance of these 331 proteins is plotted in Fig. 4C, and the 35 most abundant proteins are listed in Table 1. Among the most abundant proteins were the core components of the

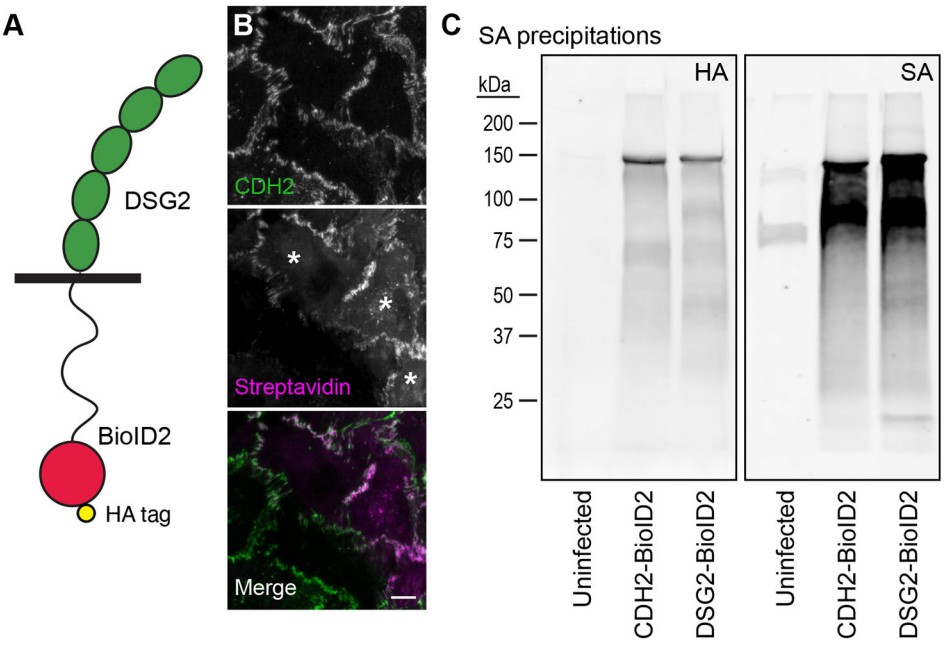

**Fig. 3. DSG2–BioID2 localizes to cell-cell contacts and labels junctional proteins.** (A) Illustration of DS2G–BioID. The BioID2 biotin ligase with a C-terminal HA tag was fused to the end of the DSG2 C-terminal tail. (B) Neonatal cardiomyocytes infected with DSG2–BioID adenovirus, incubated overnight in 50 µM biotin and fixed. Cells were stained for CDH2 (green in merge) and labeled with streptavidin conjugated to Cy3 (magenta in merge) to identify biotinylated proteins. White asterisks mark infected cells. Images representative of three experimental repeats. Scale bar: 10 µm. (C) Western blots of streptavidin precipitations from control (uninfected), CDH2–BioID2-infected or DSG2–BioID2-infected cardiomyocyte lysates. Streptavidin-precipitated material was separated by SDS-PAGE and blotted for hemagglutinin (HA, left blot) or streptavidin (SA, right blot). Blots representative of two experimental repeats.

desmosome (Fig. 4C, highlighted in green), namely, DSG2, JUP, PKP2 and DSC2. The AJ proteins (Fig. 4C, highlighted in yellow) CTTND1 (p120-catenin), CTNNB1, CTNNA3 (αT-catenin) and CDH2 were also enriched, as was the α-catenin ligand AFDN (afadin). Finally, the gap junction protein GJA1 (connexin-43, Fig. 4C, highlighted in purple) was also identified. The abundance of AJ proteins could reflect the proximity of desmosomes to AJs in the developing junctions. For example, DSG2 and PKP2 were among the most abundant hits in the CDH2 interactome (Li et al., 2019). Alternatively, AJ proteins proximal to DSG2 could indicate the intermingling of specific junctional components in hybrid junctions, as previously proposed (Borrmann et al., 2006; Franke et al., 2006). Notably, CTNNA3, considered a key component of the hybrid junction (Vite and Radice, 2014), was observed in the DSG2 interactome, whereas CTNNA1 was not.

We next classified all 325 genes in the DSG2–BioID2 interactome as belonging to one of 20 functional categories based on Uniprot, GeneCards and Entrez designations (Fig. 4D). The categories with the most hits by number were adaptor (15%), trafficking/Golgi/ER (13%), metabolic enzyme (11%) and actin-binding adaptor (11%). The top categories based on protein abundance (iBAQ) were adaptor (35%), RNA/ribosome/translation (14%) and metabolic enzyme (13%) (Fig. 4D).

We then performed enrichment analysis using Ingenuity Pathway Analysis (IPA) to determine the canonical signaling pathways represented in the DSG2 interactome. The eight most enriched pathways are depicted in Fig. 4E. Four themes emerged from IPA: (1) Rho signaling (Rho GTPase cycle), (2) trafficking (neutrophil degranulation, clathrin-mediated endocytosis), (3) cell–cell organization and signaling [cell junction organization, Sertoli cell (SC)–germ cell (GC) junction signaling, integrin signaling, and GC–SC junction) and (4) cardiomyopathy (dilated cardiomyopathy). Given the established role of the desmosome in adhesion and links to heart disease, the enrichment found for the cell–cell junction and cardiomyopathy signaling pathways was expected. The Rho GTPase and trafficking signaling pathways could reflect the prevalence of adaptor, actin-binding adaptor and trafficking proteins in the DSG2 interactome. Notably, the IPA results highlight the core adhesive and signaling functions ascribed to the desmosome.

## Comparing DSG2 and CDH2 interactomes

We compared the DSG2 interactome with the previously reported cardiomyocyte CDH2 interactome (Li et al., 2019). Note that identical MS approaches (sample size, sample processing and data analysis) were used for both studies, permitting a deep and detailed comparison. There were 172 hits shared between the two interactomes (Fig. 5A) and 153 hits unique to the DSG2 interactome. We then compared the abundance of the shared pool between the two interactomes by plotting the iBAQ abundance (Log2) of each hit in the DSG2 interactome ($x$-axis) versus the CDH2 interactome ($y$-axis) (Fig. 5B). Core AJ and desmosome proteins are marked in red. Linear regression analysis revealed a positive correlation in abundance. Note that JUP and PKP2 are heavily enriched in both interactomes. The enrichment of JUP was not surprising; it binds directly to both classical and desmosomal cadherins and is thus common to both adhesion complexes. PKP2, however, is a core component of the desmosome and does not bind directly to classical cadherins (Chen et al., 2002), although it might be recruited indirectly via proteins like CTTNA3. Nonetheless, PKP2 was highly enriched in both datasets, like JUP (Fig. 5B). CTNND1 (p120-catenin), another ARM-repeat protein like PKP2, was also enriched in both interactomes. CTNNA3 was enriched in both interactomes, whereas the more-abundant CTNNA1 was not, suggesting DSG2 preferentially associates with, or organizes near, specific AJs containing CTNNA3.

To gain further insight into potential similarities and differences between the DSG2 and CDH2 interactomes, we performed IPA enrichment analysis. We examined the DSG2 unique (i.e. proteins not shared with CHD2), CDH2 unique (i.e. proteins not shared with DSG2), and DSG2 and CDH2 (DSG2–CDH2) shared sets for enrichment in canonical signaling pathways (Fig. 5C–E). The DSG2–CDH2 shared set was enriched for Rho signaling, endocytosis and cell–cell organization/signaling (Fig. 5D). The top two canonical signaling pathways in the DSG2–CDH2 shared set – Rho GTPase cycle and clathrin-mediated endocytosis – were also the top two pathways in the CDH2 unique set (Fig. 5E). Actin cytoskeleton and muscle contraction pathways were enriched in the CDH2 set, but not in the shared or DSG2 unique sets (Fig. 5C–E), reflecting the role of the AJ in connecting the actin networks of

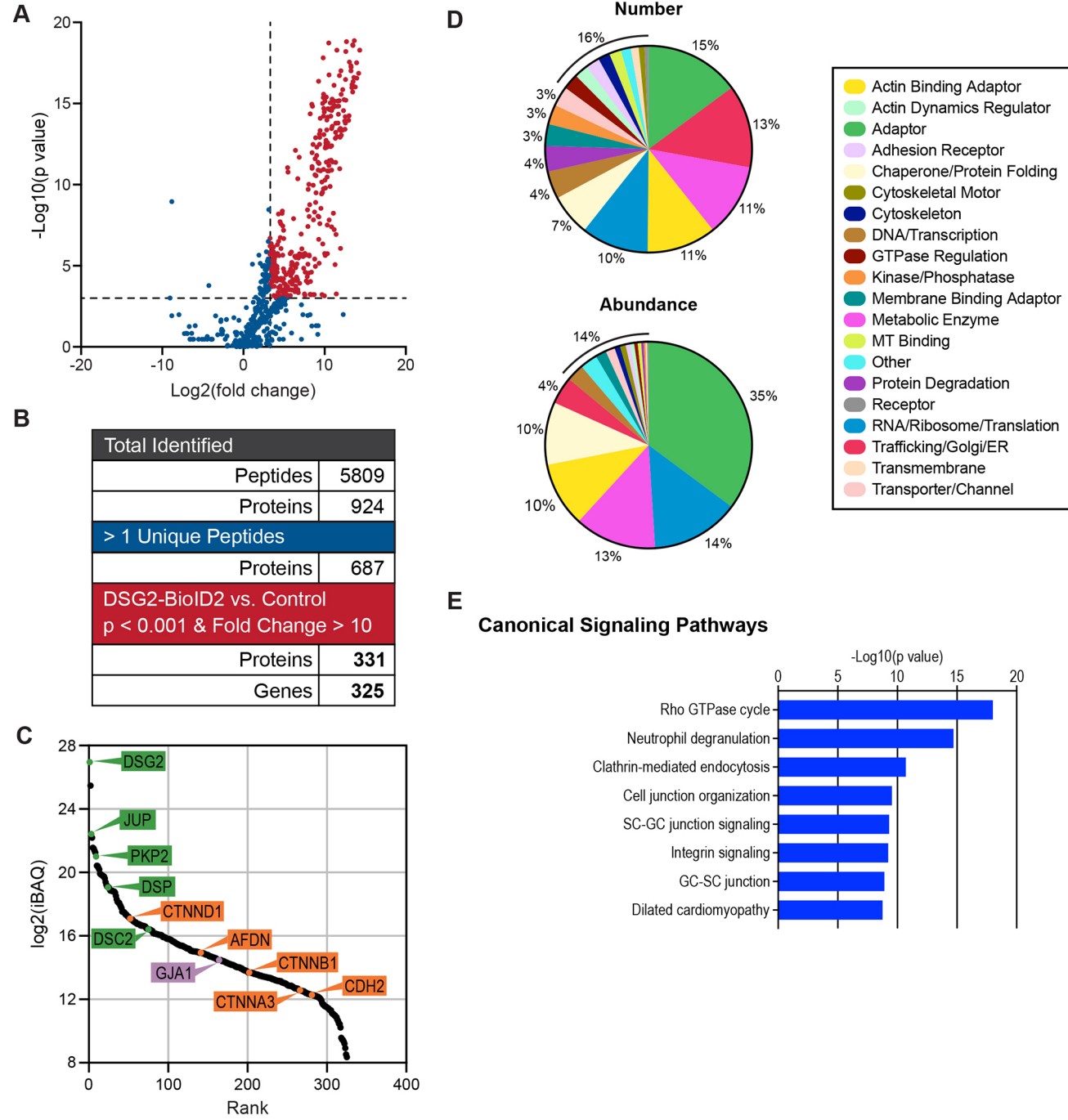

**Fig. 4. Quantitative MS identifies the DSG2 interactome.** (A) Plot of *P*-value (−log10) versus fold-change (log2) of proteins identified by quantitative mass spectrometry. Dashed lines mark *P*=0.001 (*y*-axis) and fold-change of 10 (*x*-axis). Proteins that passed selection criteria are labeled red. Plot shows pooled data from six control and six Dsg2-BioID2 replicates. (B) Summary of identified peptide and protein number at each stage of condition stringency. (C) Rank plot of abundance (iBAQ mass, log2). Proteins of interest are marked with colored circles and labeled. Circle color corresponds to the primary adhesion complex: green (desmosome), orange (adherens junction) and purple (gap junction). (D) Protein distribution by assigned category based on number (top pie chart) or abundance (iBAQ) (bottom pie chart). (E) IPA enrichment analysis of DSG2 hits in canonical signaling pathways.

adjacent cells. Finally, trafficking, mitosis and Rho signaling pathways were enriched in the DSG2 unique hits (Fig. 5C).

We then analyzed and compared the Rho GTPase cycle hits across datasets (Fig. 5F). The hits in each dataset (DSG2 unique, CDH2 unique and shared) were sorted and grouped into one for four functional categories (similar to what is shown in Fig. 4D): (1) adaptor and actin-binding, (2) protein folding and trafficking, (3) Rho GTPase regulation and (4) other. Nearly 50% of

DSG2–CDH2 shared hits were adaptor or actin-binding proteins. Adaptor and actin-binding proteins also comprised most of the CDH2 unique hits, albeit to a lesser extent (33%). In contrast, most hits associated with Rho signaling in the DSG2 unique set were trafficking and folding proteins. We speculate that differences between the DSG2 and CDH2 unique sets reflect distinct yet complementary roles in regulating cytoskeletal organization via Rho GTPase signaling.

**Table 1. Top 35 most abundant proteins in the DSG2–BioID2 interactome**

| Gene | Protein name | Category | iBAQ (log2) | % relative to DSG2 |
|---|---|---|---|---|
| Jup | Plakoglobin | Adaptor | 22.42 | 4.3 |
| Rpl23A | Ribosomal protein L23a | RNA/ribosome/translation | 22.19 | 3.7 |
| Sorbs2 | Sorbin and SH3 domain containing 2 | Adaptor | 21.55 | 2.4 |
| Gapdh | Glyceraldehyde-3-phosphate dehydrogenase | Metabolic enzyme | 21.50 | 2.3 |
| Ahnak | AHNAK nucleoprotein | Adaptor | 21.37 | 2.1 |
| Cryab | Crystallin alpha B | Chaperone/protein folding | 21.25 | 1.9 |
| Pkp2 | Plakophilin 2 | Adaptor | 21.00 | 1.6 |
| Nebl | Nebulette | Actin binding adaptor | 20.43 | 1.1 |
| Hspb1 | Heat shock protein family B member 1 | Chaperone/protein folding | 20.41 | 1.1 |
| Eef1a1 | Eukaryotic translation elongation factor 1 alpha 1 | RNA/ribosome/translation | 20.29 | 1.0 |
| Aldoa | Aldolase, fructose-bisphosphate a | Metabolic enzyme | 20.19 | 0.9 |
| Csrp3 | Cysteine and glycine rich protein 3 | Actin-binding adaptor | 19.92 | 0.8 |
| Cnn1 | Calponin 1 | Actin-binding adaptor | 19.87 | 0.7 |
| Tjp1 | Tight junction protein 1 (ZO-1) | Adaptor | 19.84 | 0.7 |
| Smpx | Small muscle protein X-linked | Other | 19.82 | 0.7 |
| Fhl1 | Four and a half LIM domains 1 | Adaptor | 19.75 | 0.7 |
| Pln | Phospholamban | Other | 19.74 | 0.7 |
| Crip2 | Cysteine rich protein 2 | DNA/transcription | 19.62 | 0.6 |
| Anxa2 | Annexin A2 | Membrane binding adaptor | 19.40 | 0.5 |
| Pkm | Pyruvate minase M1/2 | Metabolic enzyme | 19.23 | 0.5 |
| Ldha | Lactate dehydrogenase A | Metabolic enzyme | 19.10 | 0.4 |
| Dsp | Desmoplakin | Adaptor | 19.05 | 0.4 |
| Xipr1 | Xin actin binding repeat containing 1 | Actin-binding adaptor | 19.05 | 0.4 |
| Rtn4 | Reticulon 4 | Trafficking/Golgi/ER | 18.95 | 0.4 |
| Hsp90ab1 | Heat shock protein 90 alpha family Class B member 1 | Chaperone/protein folding | 18.86 | 0.4 |
| Picalm | Phosphatidylinositol binding clathrin assembly protein | Trafficking/Golgi/ER | 18.86 | 0.4 |
| Myl2 | Myosin light chain 2 | Cytoskeletal motor | 18.85 | 0.4 |
| Mlip | Muscular LMNA interacting protein | DNA/transcription | 18.81 | 0.4 |
| Tgm2 | Transglutaminase 2 | Other | 18.81 | 0.4 |
| C4of54 | Mouse RIKEN cDNA 1110002E22 | Other | 18.79 | 0.3 |
| Pakap* | Paralemmin A kinase anchor protein | Adaptor | 18.72 | 0.3 |
| Hspa8 | Heat shock protein family A (Hsp70) member 8 | Chaperone/protein folding | 18.53 | 0.3 |
| Emd | Emerin | Actin-binding adaptor | 18.30 | 0.2 |
| Ldb3 | LIM domain binding 3 | Adaptor | 18.16 | 0.2 |
| Csrp1 | Cysteine and glycine rich protein 1 | Actin-binding adaptor | 18.11 | 0.2 |

Note that DSG2 and RPS27A are excluded. *PALM2AKAP2 in humans.

## Shared desmosome components

Next, we compared the cardiomyocyte DSG2 interactome to two recent proteomic analyses of the epithelial desmosome: the DSP interactome in keratinocytes (Badu-Nkansah and Lechler, 2020), and the compiled JUP, PKP2 and DSC2a $Ca^{2+}$-dependent interactomes (which we term the 'Desmo' interactome) in MDCK cells (Fülle et al., 2024). The overlap between the three interactomes is shown as a Venn diagram in Fig. 6A. Surprisingly, only 16 proteins were shared between the multiple interactomes (listed in Table 2). Only two desmosome core proteins, DSG2 and JUP, were among the shared hits. The absence of the other core desmosome proteins is partly due to differences in plakophilin isoform expression between cell types. All interactomes contained at least one plakophilin protein: the DSG2 and Desmo interactomes shared PKP2, whereas the DSP and Desmo interactomes shared PKP1 and PKP3. DSP was observed in the DSG2 interactome but not in the Desmo combined interactome, which included proteins associated with DSC2a. In this study and the work from Fülle et al. (2024), the biotin ligase BioID2 was fused to the end of the C-terminal cytoplasmic tail in both desmosomal cadherins. However, the DSG2 cytoplasmic tail is significantly longer than the DSC2a tail and is thus more likely to promote labeling of distal desmosome proteins, such as DSP. Other proteins shared between all interactomes include the adaptor proteins CTNND1, AFDN and MSN (moesin), as well as a collection of chaperone

and trafficking proteins essential for desmosome protein folding, delivery and recycling.

We then performed IPA enrichment analysis to gain insight into the proteins shared between the interactomes. We examined the DSG2–DSP (129 shared hits), DSG2–Desmo (31 shared hits) and DSG2–DSP–Desmo (16 shared hits) sets for enrichment in canonical signaling pathways (Fig. 6B). The top pathways in the DSG2–DSP set were neutrophil degranulation, apoptotic execution phase and Rho GTPase cycle. The neutrophil degranulation and apoptotic execution phase pathways likely reflect shared trafficking and degradation proteins. The Rho GTPase cycle pathway was enriched in all shared sets (Fig. 6B), demonstrating the importance of CTNND1 and PKP proteins in regulating Rho signaling (Anastasiadis and Reynolds, 2001; Hatzfeld et al., 2014). The top pathways in the DSG2–Desmo set were Rho GTPase cycle, SC-GC junction signaling and L1CAM interactions. SG-GC junction signaling (as well as GC-SC junction signaling) pathways were enriched in all sets, reflecting the importance of desmosome proteins in the unique, but poorly understood, desmosome-like junction that connects Sertoli cells to germ cells in the testes (Lie et al., 2011). Finally, the top pathways enriched in the shared DSG2–DSP–Desmo set were L1CAM interactions, GC-SC junction signaling and Rho GTPase cycle. The L1CAM interactions pathway, also enriched in the DSG2–Desmo set, likely represents the prevalence of trafficking and adaptor proteins required for cell–cell adhesion. Overall, the analysis of

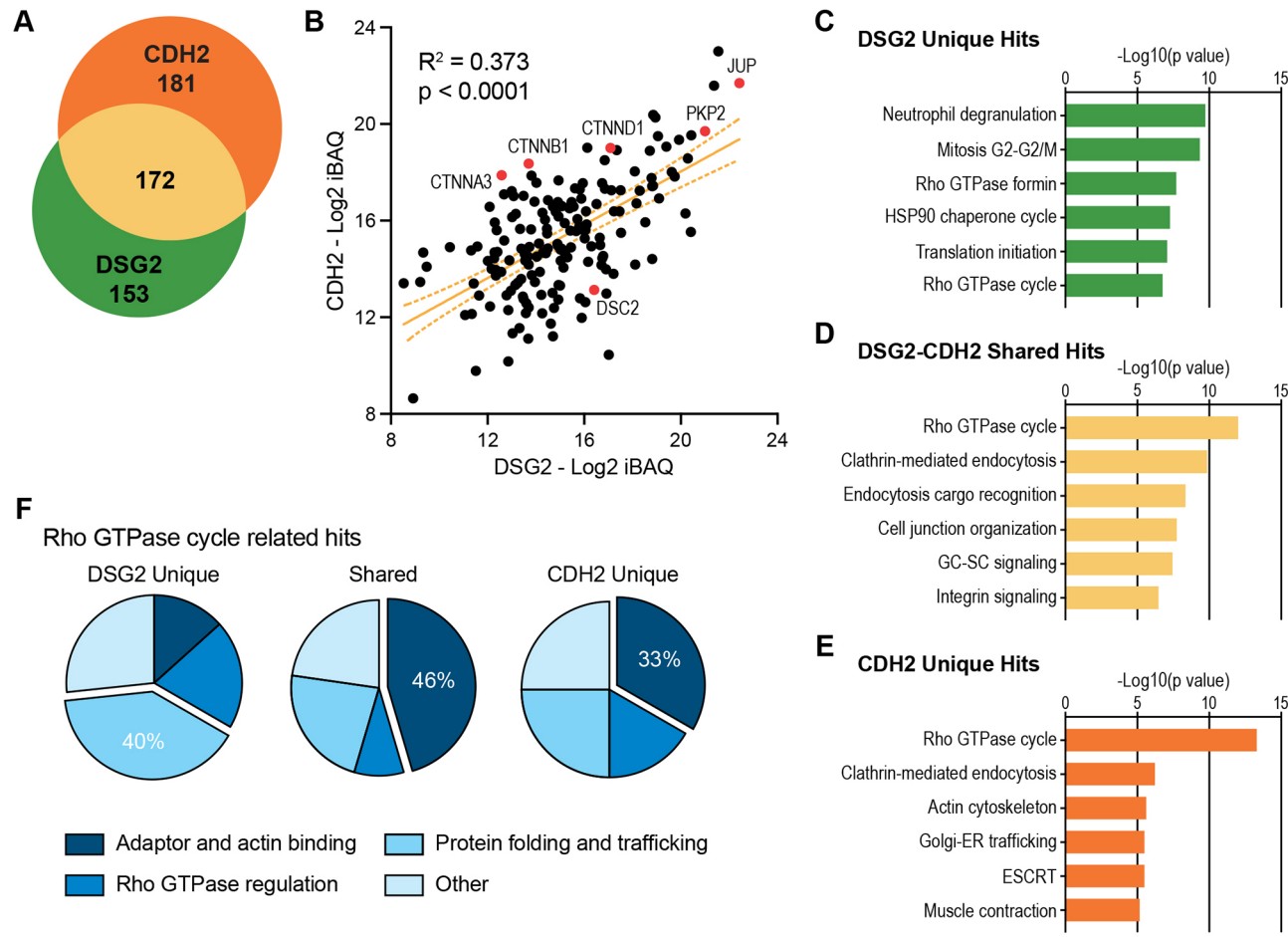

**Fig. 5. DSG2 and CDH2 interactome comparison.** (A) Venn diagram of cardiomyocyte DSG2 interactome (325 hits, green) versus cardiomyocyte CDH2 interactome (353 hits, orange). 172 hits are shared between the two interactomes (tan). Note that the original CDH2 interactome listed 354 genes, not 353. PALM2 and PALM2-AKAP were initially designated as separate genes, but they are classified as one gene (PAKAP) in mouse. (B) Scatter plot of shared hits (black dots) based on abundance (iBAQ mass, log2) in DSG2 (x-axis) and CDH2 (y-axis) interactomes. Core desmosome and adherens junction proteins are marked in red. Linear regression analysis calculated the slope (orange line), 95% confidence intervals (dashed orange lines flanking the slope) and $r^2$ value. Slope deviation from zero was analyzed for significance (P-value). (C–E) IPA enrichment analysis of DSG2 unique hits (C), DSG2–CDH2 shared hits (D) and CDH2 unique hits (E) in canonical signaling pathways. (F) Functional category breakdown of Rho GTPase cycle related hits identified in DSG2 unique (left), DSG2–CDH2 shared (middle) and CDH2 unique (right) datasets.

enriched canonical signaling pathways shared between epithelial and cardiomyocyte desmosome interactomes revealed a common reliance on adaptor, cytoskeletal remodeling and trafficking proteins that mediate intercellular adhesion across cell types.

We then analyzed hits unique to the cardiomyocyte DSG2 interactome (Fig. 6B). The top enriched pathway was dilated cardiomyopathy signaling, followed by ROBO signaling and neddylation. The prevalence of unique hits associated with dilated cardiomyopathy signaling was not surprising given the difference in cell type (cardiac versus epithelial) and the well-recognized role of the desmosome in cardiomyopathy. ROBO signaling is best known for its importance in axon guidance, but it also plays a crucial role in heart development (Zhao and Mommersteeg, 2018). Likewise, neddylation regulates protein degradation and plays a role in heart development and function (Li et al., 2020). Rho GTPase cycle signaling was also enriched in the DSG unique hits (Fig. 6B), and this was the only pathway observed in all four sets. Notably, the DSG unique hits linked to Rho GTPase cycle included the Rho GTPase CDC42 and proteins that regulate Rho GTPase activity, such as ARHGAP1 (Rho GTPase activating protein 1), ARHGDIA (Rho GDP dissociation inhibitor α) and RASAL2 (RAS protein activator-

like 2). Rho signaling unique to the DSG2 interactome could reflect a cardiomyocyte-specific repertoire of Rho GTPase regulatory proteins necessitated by the unique cytoskeletal organization and physical demands of cardiomyocyte cell–cell adhesion.

### DSG2 interactome protein network
We connected and organized the DSG2 interactome to gain a deeper understanding of the molecular complexes assembled at cardiomyocyte desmosomes and how these complexes might differ from those at cardiomyocyte AJs (Fig. 7A). We classified all interactors as either unique to DSG2 (green circles) or shared between DSG2 and CDH2 (yellow circles). The top 35 hits in the DSG2 interactome are circled in orange. We then constructed a hierarchical classification with DSG2 at the top (see figure legend and Materials and Methods for details). We separated the interacting network into four tiers: 11 primary, 138 secondary, 127 tertiary and 10 quaternary (Fig. 7A). All protein–protein interactions were based on published experimental data. A total of 39 of the DSG2–BioID hits could not be connected to any protein in the network (listed in Table S2). We then calculated the percentage distribution of DSG2 and DSG2–CDH2 hits within each tier (Fig. 7B). The percentage of

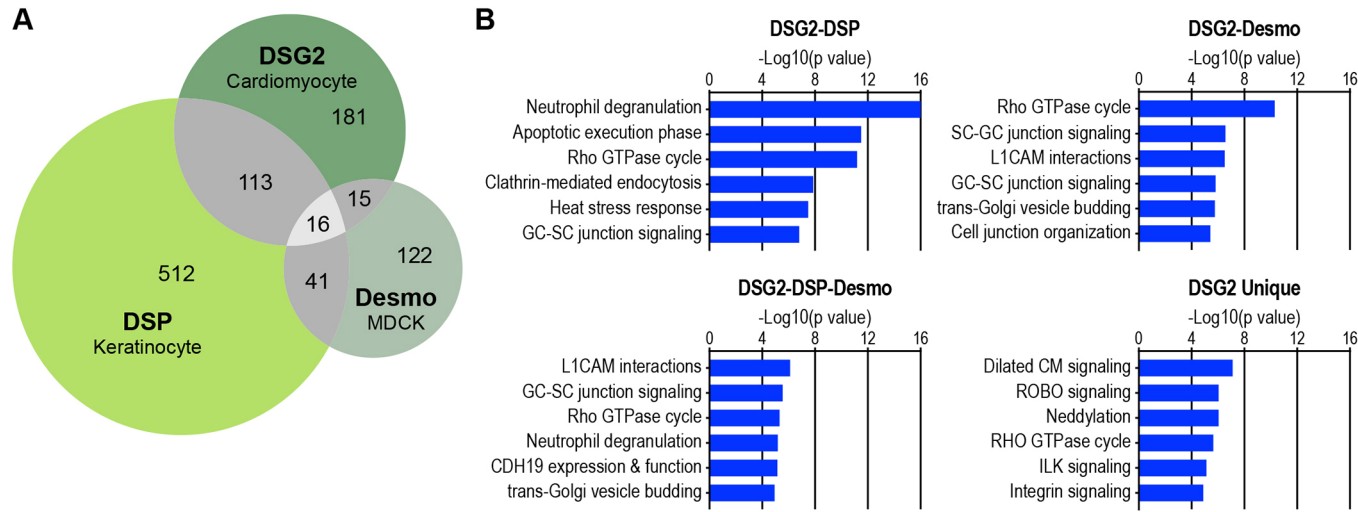

**Fig. 6. Desmosome interactome comparisons.** (A) Venn diagram of cardiomyocyte DSG2 interactome (325 hits), DSP interactome in keratinocytes (682 hits), and the combined JUP, PKP2 and DSC2a Ca$^{2+}$-dependent interactome in MDCK cells (labeled Desmo for desmosome; 194 hits in total); 16 proteins are shared between the interactomes. (B) IPA enrichment analysis of DSG2–DSP shared hits (top left), DSG2–Desmo shared hits (top right), DSG2–DSP–Desmo shared hits (bottom left) and DSG2 unique hits (bottom right) in canonical signaling pathways.

DSG2–CDH2 shared hits was highest in the primary (73%), tertiary (61%) and quaternary tiers (80%), but noticeably reduced in the secondary tier (38%). The high percentage of shared hits in the primary tier likely reflects shared core components (e.g. JUP, CTNNA3 and PKP2). In contrast, the large percentage of shared hits in the tertiary and quaternary tiers could indicate shared ancillary proteins required for cadherin trafficking, stability and degradation, among other functions. DSG2-only hits dominated the secondary tier, and we speculate that these include specific, specialized interactors recruited to the desmosome core complex to promote adhesion and initiate signaling.

We then examined the direct connections to the desmosome complex proteins DSG2, DSC2, JUP, PKP2 and DSP (Fig. S1). Desmosome complex proteins interacted directly with at least 50 peripheral proteins (i.e. non-desmosomal proteins) in the DSG2 interactome, nearly 20% of the mapped network in Fig. 7. Among these 50 hits, 12 bound more than one desmosomal protein (e.g. SYNPO associates with JUP and DSG2), and 24 were shared hits with CDH2 (colored yellow, Fig. S1). The desmosome protein with

**Table 2. DSG2–DSP–Desmo interactomes shared hits**

| Gene | Protein | Category |
| --- | --- | --- |
| Dsg2 | Desmoglein 2 | Adhesion receptor |
| Jup | Plakoglobin | Adaptor |
| Ctnnd1 | p120-catenin | Adaptor |
| Afdn | Afadin | Actin-binding adaptor |
| Msn | Moesin | Actin-binding adaptor |
| Sptan1 | Spectrin alpha, non-erythrocytic 1 | Actin-binding adaptor |
| Cryab | Crystallin alpha B | Chaperone/protein folding |
| Dnajb1 | DnaJ heat shock protein family (Hsp40) member B1 | Chaperone/protein folding |
| Hspa8 | Heat shock protein family A (Hsp70) member 8 | Chaperone/protein folding |
| Clint1 | Clathrin interactor I | Trafficking/Golgi/ER |
| Cltc | Clathrin heavy chain | Trafficking/Golgi/ER |
| Nsfl1c | NSFL1 cofactor | Trafficking/Golgi/ER |
| Snap29 | Synaptosome associated protein 29 | Trafficking/Golgi/ER |
| Psmc2 | Proteasome 26S subunit, ATPase 2 | Protein degradation |
| Rpl3 | Ribosomal protein L3 | RNA/ribosome/translation |
| Slc3a2 | Solute carrier family 3 member 2 | Transporter/channel |

the most interactions was JUP, which associates with 36 of the 50 hits (Fig. S1). The large number of direct associations with individual desmosomal proteins underscores the fundamental role of the desmosome complex in adhesion, scaffolding and signaling.

## PKP2 recruitment to cell–cell contacts is regulated by tension

The interactome network analysis revealed that 73% of the proteins thought to interact directly with DSG2 (primary tier) are also associated with CDH2, although most do not bind CDH2 directly. One notable exception is JUP, which binds with high affinity to the tails of both classical cadherins and desmosomal cadherins. Not surprisingly, JUP was one of the most abundant hits in both the DSG2 and CDH2 interactomes (Figs 4C and 5B, Table 1; Li et al., 2019). Nearly as abundant in both interactomes was PKP2 (Figs 4C and 5B, Table 1; Li et al., 2019). PKP2 is primarily regarded as a desmosome protein, although it has been shown to associate with the AJ in cardiac and non-muscle cells (Borrmann et al., 2006; Chen et al., 2002; Franke et al., 2006). We decided to examine the localization of endogenous PKP2 in neonatal cardiomyocytes and compare it with that of other cell–cell junction proteins. Confluent cultures of cardiomyocytes were fixed and stained for PKP2 and CTNNB1, CTNNA1, CTNNA3, JUP, afadin and GJA1 (Fig. 8A–F). PKP2 was enriched at cell–cell contacts, and the localization looked nearly identical to an AJ protein. Pearson's colocalization analysis of PKP2 with other junctional proteins confirmed this. We measured high Pearson's correlation coefficients for PKP2 with CTNNB1, JUP and CTNNA1 (Fig. 8G). We also calculated high, although significantly lower, Pearson's coefficients for PKP2 with CTNNA3 and AFDN (Fig. 8G). In contrast, a significantly lower correlation was noted between PKP2 and GJA1 (Fig. 8G). Based on these results and our MS data, we suggest that PKP2 should be considered a component of both the desmosome and the AJ in cardiomyocytes, similar to JUP.

It has been proposed that PKP is recruited to the middle (M) domain of CTNNA1 to form a hybrid junction in the heart (Goossens et al., 2007; Li et al., 2012). The CTNNA3 M-domain, like that of CTNNA1, is mechanosensitive in that applied load opens the domain to reveal a binding site for vinculin (Pang et al., 2019). Although there are differences in M-region regulation between CTNNA1 and

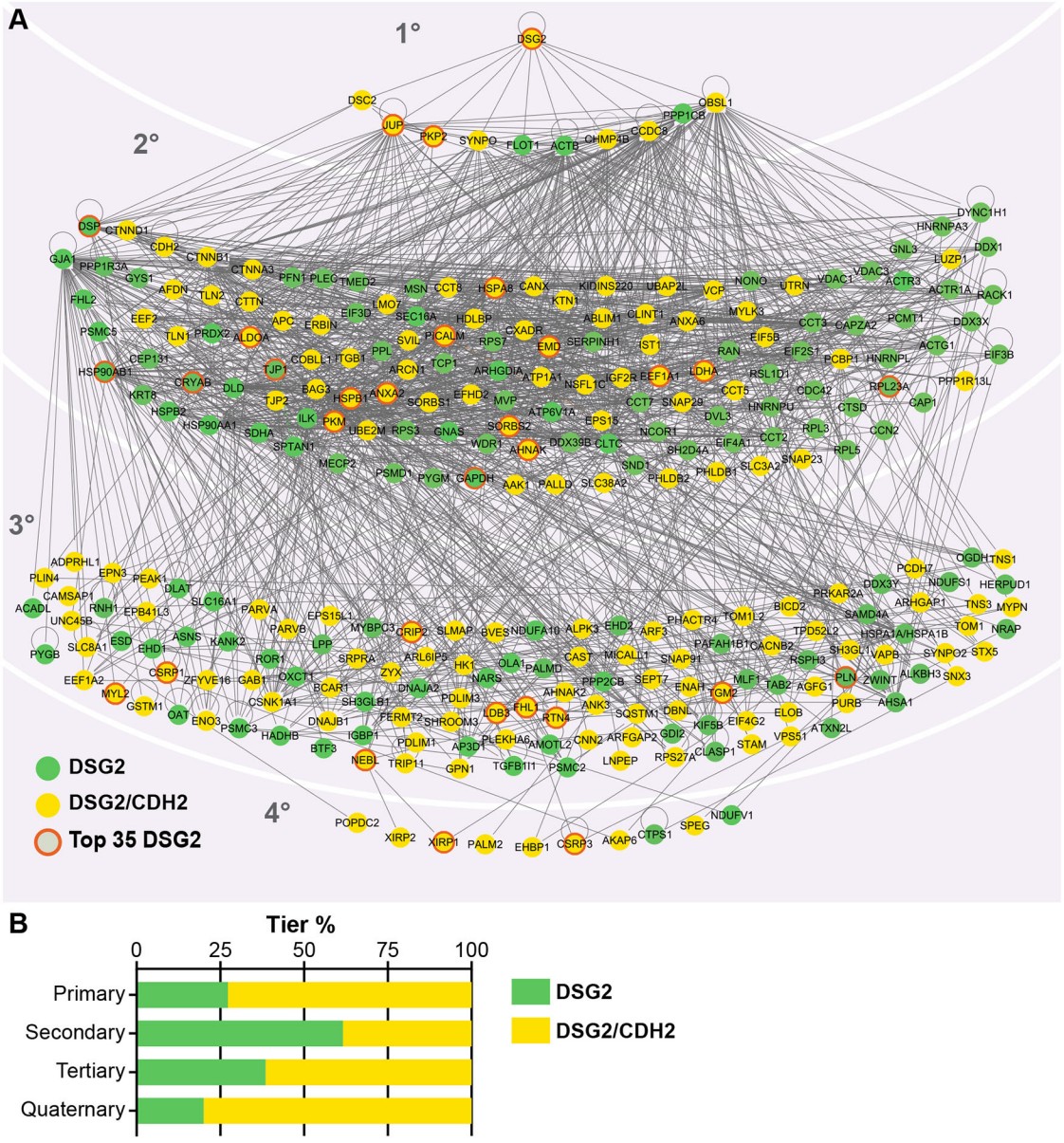

**Fig. 7. Cardiomyocyte DSG2 interactome network.** (A) DSG2 interactome interaction network organized into four tiers based on IPA. Published experimental data support all protein–protein interactions. Hierarchical classification was performed manually around DSG2. Primary interactors bind DSG2 directly. Secondary interactors bind to primary interactors, but not to DSG2. Tertiary interactors bind secondary interactors, and quaternary interactors bind tertiary interactors. Green circles signify hits unique to the DSG2 interactome; yellow circles mark hits shared with the CDH2 interactome. The 35 most abundant DSG2 interactome proteins (Table 1) are outlined in orange; 39 proteins not connected to the network are listed in Table S2. (B) Percentage distribution of DSG2 unique (green bar) and DSG2–CDH2 shared (yellow bar) hits within each tier.

CTNNA3 (Choi et al., 2012; Heier et al., 2021), both are force-sensitive, and mechanical load is required to promote vinculin binding (Le et al., 2019; Maki et al., 2018; Pang et al., 2019; Seddiki et al., 2018; Yao et al., 2014; Yonemura et al., 2010). We questioned whether PKP2 recruitment to cardiomyocyte cell–cell junctions was dependent on tension. Cultured cardiomyocytes were treated with dimethyl sulfoxide (DMSO) alone or DMSO plus 100 µM blebbistatin to suppress myosin activity for 1 h. Cells were then fixed and stained for PKP2, JUP and F-actin (Fig. 8H–J). PKP2 at cell–cell junctions was significantly reduced in blebbistatin-treated cells after 1 h (Fig. 8I–K; estimation plot in Fig. S2). In contrast, blebbistatin treatment caused a small, but significant, increase in JUP at cell–cell contacts (Fig. S2). These results are consistent with a requirement for tension to recruit PKP2, possibly to α-catenin.

Although PKP2 junction levels decreased after blebbistatin treatment, a pool of PKP2 remained localized to cell–cell junctions. We speculate that this tension-insensitive pool of PKP2 might be directly bound to desmosomal proteins (DSG2, DSC2, JUP and DSP). Together, these results show that, in neonatal cardiomyocytes, PKP2 is a ubiquitous junction protein recruited, in part, through tension.

## DISCUSSION

We describe the first detailed analysis of the DSG2 interactome in cardiomyocytes. Our results provide insight into DSG2 molecular associations in developing cardiomyocyte cell–cell adhesions. We uncovered an extensive set of shared and DSG2-specific interactions that highlight the unique mechanical environment of

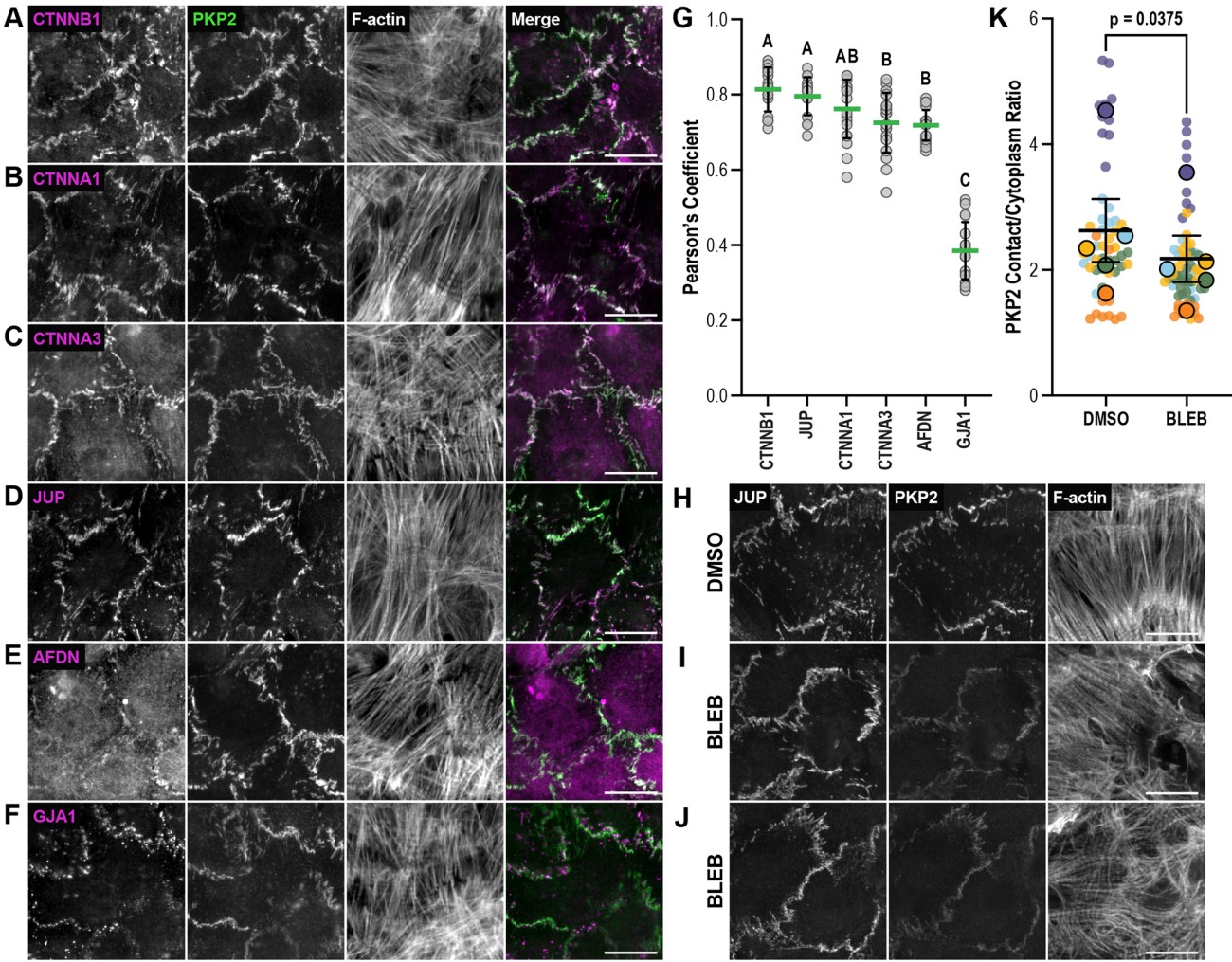

**Fig. 8. PKP2 is recruited to cardiomyocyte AJs under tension.** (A–F) Cardiomyocytes were cultured for 72–96 h, fixed, and stained for PKP2, F-actin and βcat (A), αEcat (B), αTcat (C), JUP (D), AFDN (E) or Cx43 (F). All images are maximum projections of 1 µm z-stacks. Merge images show overlap between PKP2 (green) and adhesion protein (magenta) channels. (G) Pearson's coefficient was calculated between PKP2 and individual adhesion protein channels at cell–cell contacts. Scatter plot of all data points shown, 30 images total from three biological replicates. The thick horizontal green line marks the mean, and the black error bars define the s.d. Each letter display defines coefficients that are statistically indistinguishable ($P<0.05$, one-way ANOVA with Tukey's multiple comparisons). (H–J) Cardiomyocytes were cultured for 72 h, incubated for 1 h in DMSO (control, H) or 100 µM blebbistatin (I,J), fixed, and stained for JUP, PKP2 and F-actin. (K) PKP2 average fluorescence signal intensity at cell–cell contacts divided by the average intensity in the cytoplasm (PKP contact/cytoplasm ratio) in cells treated with DMSO or blebbistatin. Data are presented as a SuperPlot (Lord et al., 2020), where individual measured PKP contact/cytoplasm fluorescence ratios (data points) from one biological replicate are plotted as circles, and each replicate is assigned a unique color ($n=5$). The mean PKP contact/cytoplasm fluorescence ratio from each replicate is overlaid on the individual data points as a larger, color-matched circle outlined in black. The mean±s.e.m. of the five replicates are shown as black bars. Blebbistatin treatment caused a significant decrease in PKP2 enrichment at cell-cell contacts ($P=0.0375$, paired two-tailed $t$-test). Scale bars: 20 µm.

cardiomyocyte intercellular adhesions and the interplay between the AJ and desmosome.

## The DSG2 interactome

This work builds off our past study of the CDH2 interactome in cardiomyocytes (Li et al., 2019) and complements more recent studies of the desmosome proteome in epithelia (Badu-Nkansah and Lechler, 2020; Fülle et al., 2024; Hegazy et al., 2022) as well as past studies of the epithelial AJ CDH1 (E-cadherin) interactome (Guo et al., 2014; Shafraz et al., 2020; Van Itallie et al., 2014). Proteomic studies to define the desmosome interactome in epithelia used similar biotinylation-based approaches to identify the interactomes of DSP (Badu-Nkansah and Lechler, 2020) and DSG1 (Hegazy et al., 2022) in keratinocytes, and DSC2a, JUP and PKP2a in MDCK cells (Fülle et al., 2024). Here, we add the

DSG2 interactome to these datasets, expanding our appreciation of desmosomes by defining DSG2 interactions in the context of an adhesion system under load with unique organizational and mechanical demands.

Our quantitative analysis revealed that the most abundant proteins associated with DSG2 (Fig. 4C, Table 1) are the foundational desmosome proteins – JUP, PKP2, DSP and DSC2. However, the limited number of shared hits beyond the desmosome core between the multiple desmosomal protein interactomes was somewhat surprising. We speculate that the dearth of shared proteins reflects cell-specific or tension-dependent requirements for desmosome adhesion, trafficking, turnover and/or cytoskeletal integration. Nonetheless, a complex network of adhesion protein interactions has emerged, varying between adhesion types, adhesion states and tissue sources. Defining how these interactions regulate the

mechanical and signaling properties of the desmosome and AJ, as well as potential hybrid and transitory adhesion complexes, will be the focus of future work.

The CDH2 interactome in neonatal cardiomyocytes includes many specific adaptor and linker proteins that likely strengthen cytoskeletal connections to cardiac AJs under increased load. We presume the cardiomyocyte desmosome experiences high tension. Consistent with this, a DSG2 FRET-based tension sensor has been used to show that DSG2 is under greater load in contracting cardiomyocytes than in MDCK cells (Baddam et al., 2018). In addition, we have recently shown that the DSP conformation is altered by actomyosin force in cardiomyocytes (Dong et al., 2025). We found that adaptor and actin-binding proteins comprise 26% of the DSG2 interactome by number and 45% by abundance (Fig. 4D). We conclude that the cardiomyocyte desmosome, like the cardiomyocyte AJ, recruits a bevy of secondary scaffolding, adaptor and/or cytoskeletal proteins that could assist in mechanical adhesion.

Within the DSG2 interactome, more than 100 proteins have been shown to bind either IFs or actin. Furthermore, some proteins have been shown to bind both actin and IFs or link the two cytoskeletons, including nebulette (NEBL), synemin (SYNM) and plectin (PLEC) (Hernandez et al., 2016; Prechova et al., 2022; Russell, 2020). These proteins could mediate multiple cytoskeletal attachments to the desmosome, similar to those at the hybrid junction (discussed below). In addition, three plakin proteins were identified – DSP, PLEC (mentioned above) and periplakin (PPL). Plakin proteins are large, versatile linker proteins that connect IFs to junctional complexes (Leung et al., 2002), and mutations in PLEC, as for DSP, are associated with cardiomyopathy and cardiac disease (Bolling et al., 2010; Villa et al., 2015). Actin and IF integration at cardiomyocyte cell–cell junctions is poorly understood, and how the adaptor proteins function with DSG2 to regulate cardiomyocyte adhesion will be addressed in future studies.

### DSG2 is the primary mechanical link to the cytoskeleton

DSG and DSC function together to mediate intercellular adhesion in mature desmosomes. Although homotypic *trans* interactions (i.e. DSG–DSG) can occur and might have a physiological role, heterotypic *trans* interactions (i.e. DSG–DSC) are much stronger *in vitro* and likely represent the dominant interface *in vivo* (Harrison et al., 2016). DSG cadherins possess a long cytoplasmic tail with established binding sites for JUP and PKP, as well as additional ligands (Bharathan et al., 2024). DSC cadherins, on the other hand, exist as two spliceforms – an 'a' isoform with a cytoplasmic tail that binds JUP and PKP proteins, and a 'b' isoform with a truncated tail that lacks the complete CBD and does not bind JUP. In mouse neonatal cardiomyocytes and heart tissue, we found that both Dsc2a and Dsc2b were expressed (Fig. 1C). A similar expression pattern was observed in the adult human heart (De Bortoli et al., 2010). Given that the partial deletion of the CBD in DSCb removes much of the predicted JUP binding interface (Pasani et al., 2024) and prevents JUP binding in cells (Troyanovsky et al., 1994), heterotypic *trans* interactions between DSG2 and DSC2b would potentially create a mechanical imbalance within the intercellular complex based on the asymmetric recruitment of JUP. Such an imbalance could be advantageous in contractile tissues, enabling desmosomes to absorb mechanical strain. Likewise, an imbalance could mediate desmosome remodeling during cardiomyocyte maturation (Guo and Pu, 2020). Alternatively, an imbalance could be remedied by *cis* interactions (extracellular or intracellular) between neighboring desmosome complexes, or through the ability of DSC2b to connect to IFs through PKP2 recruitment. Although there are

notable differences in ligand binding between PKP homologs (Bonne et al., 2003), structural modeling suggests that PKP1 binds a region of DSC1 tail that is also present in DSC1b (Pasani et al., 2024). Finally, DSC2a could also provide sufficient mechanical linkage even at substoichiometric levels relative to DSC2b. However, DSC2b can compensate for DSC2 variants that prevent DSC2a tail expression in humans, suggesting that DSC2b is sufficient for desmosome function in the heart (De Bortoli et al., 2010).

Loss of DSG2 function in the murine heart prevents desmosome formation, induces cardiomyocyte necrosis and fibrosis, and causes cardiomyopathy (Kant et al., 2015). Note that DSG2 expression in the rat heart is roughly five times that of DSC2, suggesting that DSG2 might possess functions independently of DSC2. Consistent with this, there are AC-linked mutations in surface residues of the DSG2 EC3 domain, well outside the *trans*-binding interface (Harrison et al., 2016). Furthermore, DSG2 has been shown to engage in $Ca^{2+}$-dependent *trans* interactions with CDH1 (E-cadherin) and CDH2 (Fuchs et al., 2022), as well as *cis* interactions with CDH1 (Shafraz et al., 2018). Much remains to be learned about the DSG2–DSC2 adhesion complex and the roles of both DSG2 and DSC2 in the heart.

### PKP2 is recruited to AJs and desmosomes in the heart

JUP and PKP2 were among the most abundant hits in both the CDH2 and DSG2 interactomes. JUP is a homolog of β-catenin and binds directly to the cytoplasmic tails of classical cadherins and desmosomal cadherins. PKP2, however, binds directly to JUP, DSP, DSG and DSC, and is primarily considered a core component of the desmosome. There is, however, evidence for PKP2 associations outside the desmosome, and it has been suggested that PKP proteins coordinate AJ and desmosome assembly (Bass-Zubek et al., 2009). Overexpressed PKP2 in keratinocytes localized to cell–cell contacts in a continuous, rather than punctate, pattern, suggesting membrane recruitment independent of the desmosome. When overexpressed in COS cells, β-catenin co-precipitates with PKP2 (Chen et al., 2002). Likewise, endogenous PKP2 co-precipitates with N-cadherin from bovine heart lysates (Borrmann et al., 2006). Furthermore, immunoelectron microscopy studies of mammalian heart tissue have revealed PKP2 localization along the entire ICD (Franke et al., 2006). The intermingling of desmosomal and AJ components along the ICD, as observed by Franke and colleagues, led to the concept of the area composita, or hybrid junction. We observed a continuous pattern of PKP2 localization at cardiomyocyte cell–cell contacts. PKP2 membrane recruitment was indistinguishable from that of JUP, CTTNA1, CTNNA3 and CTNNB1, as reflected in the high Pearson's colocalization coefficients (Fig. 6G). This localization pattern is consistent with past immunoelectron imaging of adult mammalian hearts and immunofluorescent studies of cultured rat neonatal cardiomyocytes (Franke et al., 2006, 2007). Combined with our current and past MS results, we propose that PKP2 is a component of the cardiomyocyte desmosome and AJ. As noted above, neonatal cardiomyocytes do not form mature ICDs, and we do not consider these mixed junctions true area composita. However, Franke and colleagues showed that PKP2 is an integral component of both CDH2 and DSG2 complexes and suggest that component mixing is an intrinsic property of both developing and mature cardiomyocyte cell–cell contacts (Borrmann et al., 2006; Franke et al., 2006, 2007). This wider, desmosome-independent recruitment of PKP2 to cardiomyocyte adhesions is essential when considering PKP2 function in cardiac health and disease.

Finally, we showed that PKP2 recruitment to junctions is dependent on tension. Force is required to unfurl the mechanosensitive α-catenin M-region and reveal binding sites for ligands such as VCL, and we

Journal of Cell Science

speculate that force is also necessary to promote PKP2 binding to α-catenin. Note, however, that a pool of PKP2 remained membrane bound in the absence of tension. This tension-independent PKP2 pool could reflect head-domain binding to other ligands (e.g. JUP) and/or binding of the ARM repeats to the DSG2 or DSC2b tails.

### Limitations

While desmosomes form between cultured neonatal cardiomyocytes (Fig. 1B), the cell-cell junctions are not mature in this model. We often refer to these junctions as pre-ICDs to distinguish them from mature ICDs, which are only observed *in vivo*. Thus, these newly formed desmosomes may lack elements of specialization required in mature cardiac tissue. Desmosome assembly is also more complicated than AJ assembly in epithelia (Beggs et al., 2022; Bharathan et al., 2024; Kowalczyk and Green, 2013; Nekrasova and Green, 2013), though desmosome maturation is poorly defined in CMs. Finally, between labeling time length (24 h) and variability in adhesion maturation within the culture, biotin labeling 'captures' the DSG2 interactome at multiple stages of desmosome formation. Future experiments with faster ligases (TurboID; Branon et al., 2018), different culture conditions, and *in vivo* cardiac labeling could add deeper insight into desmosome organization and function.

### Summary

Building on our previous work on the cardiomyocyte CDH2 interactome, we have identified over 500 proteins that comprise the core of the cardiac AJ and desmosome. This interconnected array of proteins regulates all aspects of cadherin biology, including trafficking, organization, adhesion, post-translational modification and downstream signaling. Defining how these cadherin-associated proteins function, individually and collectively, to regulate desmosome, AJ and ICD biology in the heart remains the next challenge.

## MATERIALS AND METHODS
### Plasmids

To create the mouse Dsc2b and Dsg2 (mDsc2b and mDsg2) constructs, RNA was first isolated and purified from adult mouse heart using the RNeasy Fibrous Tissue Mini kit (Qiagen) and reverse transcribed to create cDNA using the Transcriptor High Fidelity cDNA Synthesis Kit (Roche). Gene-specific primers were designed against the 5′ and 3′ ends of each gene to generate full-length clones by PCR. PCR products were cloned directly into pCR-BluntII-TOPO (Thermo Fisher Scientific) and sequence verified. mDsg2 and mDsc2b CDS were cloned into pEGFP-N1 (Takara Bio) to create EGFP fusions. Human desmoplakin-GFP plasmid was from Addgene (Addgene plasmid #32227; RRID:Addgene_32227, deposited by Kathleen Green); human Dsc2a was cloned from Desmocollin 2-myc (Addgene plasmid #32233; RRID:Addgene_32233, deposited by Kathleen Green) into pEGFP-N1 and human PKP2a was cloned from Plakophilin 2a-Flag (Addgene plasmid #32230; RRID:Addgene_32230, deposited by Kathleen Green) into pEGFP-C1. All constructs were sequence verified.

### Antibodies

Primary antibodies used for immunostaining were: anti-αE-catenin (1:100, Enzo Life Science, ALX-804-101-C100), anti-αT-catenin (1:100, rat, a kind gift from Jolanda Van Hengel, Ghent University), anti-β-catenin (1:100, Cell Signaling, D10A8), anti-I-afadin (1:500, Sigma, A0349), anti-γ-catenin (1:100, Cell Signaling Technology, 2309), anti-connexin-43 (1:100, Proteintech, 15386-1-AP) and anti-plakophilin 2 (1:10, Progen, 651101). Streptavidin-Cy3 (1:300, Jackson Immunoresearch, 016-160-084) was used to label biotinylated proteins. Secondary antibodies used were goat anti-mouse- or anti-rabbit-IgG labeled with Alexa Fluor 647 (1:250, Thermo Fisher Scientific).

### Dsc2 isoform analysis

RNA was isolated from 3-week-old mouse hearts or cultured neonatal cardiomyocytes (see below) and reverse-transcribed to create cDNA using the High Capacity RNA to cDNA Kit (Thermo Fisher Scientific). Purified cDNA was then used as the template for PCR to amplify the Dsc2 region surrounding exon 16 using two different primer sets (1F, 5′-GGCTGGCTATCATCACCATA-3′; 1R, 5′-GCTTCTGGTTGTC-ATCCGTA-3′; 2F, 5′-GAGGTGGACAACTACAGACA-3′; 2R, 5′-CTTGAGCAAGCTTCTGGTTG-3′).

### Cardiomyocyte isolation and culture

All animal work was approved by the University of Pittsburgh Division of Laboratory Animal Resources. Primary cardiomyocytes were isolated from Swiss Webster mice as described previously (Ehler et al., 2013). For protein isolation, cardiomyocytes were plated onto 35 mm dishes ($10^6$ cells/dish) coated with collagen type I (Millipore). For immunostaining, cardiomyocytes were plated onto 35 mm MatTek dishes with 10 mm insets coated with collage type I. Cardiomyocytes were plated in plating medium: 65% high glucose DMEM (Thermo Fisher Scientific), 19% M-199 (Thermo Fisher Scientific), 10% horse serum (Thermo Fisher Scientific), 5% FBS (Atlanta Biologicals) and 1% penicillin-streptomycin (Thermo Fisher Scientific). Medium was replaced 16 h after plating with maintenance medium: 78% high glucose DMEM, 17% M-199, 4% horse serum, 1% penicillin-streptomyocin, 1 μM AraC (Sigma) and 1 μM isoproternol (Sigma). Cells were cultured in maintenance medium for 2–4 days until lysis or fixation.

### Immunostaining and confocal microscopy

Cells were fixed in 4% EM grade paraformaldehyde in PHEM buffer (60 mM PIPES pH 7.0, 25 mM HEPES pH 7.0, 10 mM EGTA pH 8.0, 2 mM $MgCl_2$ and 0.12 M sucrose) or PHM (no EGTA) buffer for 10 min, washed twice with PBS, and then stored at 4°C until staining. Cells were permeabilized with 0.2% Triton X-100 in PBS for 5 min and washed twice with PBS. Cells were then blocked for 1 h at room temperature in PBS+10% BSA (Sigma), washed twice in PBS, incubated with primary antibodies in PBS+1% BSA for 1 h at room temperature, washed twice in PBS, incubated with secondary antibodies in PBS+1% for 1 h at room temperature, washed twice in PBS and then mounted in ProLong Glass (Thermo Fisher Scientific). All samples were cured for at least 24 h before imaging.

For blebbistatin experiments, cardiomyocytes (72 h post plating) were treated with 100 μM blebbistatin in DMSO or DMSO alone for 1 h. Cells were incubated at 37°C during treatment. After incubation, cells were fixed and labeled as described.

Cells were imaged on a Nikon Eclipse Ti inverted microscope equipped with a Prairie swept-field confocal scanner, an Agilent monolithic laser launch, and an Andor iXon3 camera. The microscope was run by NIS Elements (Nikon) imaging software.

### Image analysis

To measure colocalization between PKP2 and adhesion proteins in cardiomyocytes, a 1-μm maximum projection containing cell–cell contacts was generated for analysis in ImageJ (NIH). IsoJ Dark thresholding was then used to create a mask of the PKP2 staining channel to define the region of analysis (cell–cell contacts). A Pearson's coefficient was calculated between PKP2 and an individual adhesion protein channel within the masked region, and the results were plotted in Prism (GraphPad). One-way ANOVA with Tukey's multiple comparisons was used to calculate significance.

To measure average PKP2 fluorescence signal intensity at cell–cell contacts in the blebbistatin experiments, a single plane was selected from the z-stack where the cell–cell contacts were most in focus for analysis in ImageJ (NIH). IsoJ Dark thresholding was used to create a mask of the JUP channel to define the region of analysis (cell–cell contacts). The average PKP2 and JUP signal intensities were measured in the masked region and divided by the average PKP2 and JUP cytoplasmic intensities, respectively, to calculate PKP2 or JUP contact-to-cytoplasmic ratios. Data were graphed as SuperPlots (Lord et al., 2020) in Prism (GraphPad). Statistical significance was determined using a paired two-tailed t-test.

## FRAP experiments

FRAP experiments were conducted on a Nikon swept-field confocal microscope (described above) outfitted with a Tokai Hit cell incubator and a Bruker miniscanner. Actively contracting cells were maintained at 37°C in a humidified atmosphere containing 5% $CO_2$. User-defined regions of 2–3 µm in width along cell–cell contacts were bleached with a 405 nm laser, and recovery images were collected every 10 s for 10 min. FRAP data were quantified in ImageJ (NIH). Individual recovery data were analyzed in Prism (GraphPad) using nonlinear regression (two-phase association) with the following constraints: the plateau must be between 0 and 1, and the $y$-intercept must be 0. Following analysis, individual recovery plots were excluded if the $r^2$ value was below 0.85. The calculated plateau was then compared to the average of the last four data points in the set. If the difference between the calculated plateau and the average was greater than 0.25, the recovery plot was excluded. The remaining individual recovery plots were plotted and analyzed in Prism. The plotted data were fit to a two-phase association curve to determine the mobile fraction and halftime of recovery. FRAP recovery plots represent data from at least three separate transfections from at least two independent cell preps.

## Electron microscopy

Cardiomyocytes were cultured on collagen-coated MatTek dishes and fixed as described above. After fixation and washing, cells were incubated with 1% OsO4 for 1 h. After several PBS washes, dishes were dehydrated through a graded series of ethanol from 30% to 100%, then infiltrated for 1 h in Polybed 812 epoxy resin (Polysciences). After several changes of 100% resin over 24 h, cells were embedded in inverted Beem capsules, cured at 37°C overnight, and then hardened for 2 days at 65°C. Blocks were removed from the glass dish via a freeze-thaw method by alternating liquid nitrogen and 100°C water. Ultrathin (60 nm) sections were collected on to 200-mesh copper grids, stained with 2% uranyl acetate in 50% methanol for 10 min and 1% lead citrate for 7 min. Samples were photographed with a JEOL JEM 1400 PLUS transmission electron microscope at 80 kV with a Hamamatsu ORCA-HR side mount camera.

## Adenovirus production

The mDsg2 was cloned into MCS-BioID2-HA (Addgene plasmid # 74224; RRID:Addgene_74224; deposited by Kyle Roux) to fuse BioID2 to the C-terminal tail of DSG2. The Dsg2–BioID2 fusion was then subcloned into pAdTrack-CMV (Addgene plasmid #16405; RRID:Addgene_16405; deposited by Bert Vogelstein) (He et al., 1998). Recombinant adenovirus was produced by transforming the pAdTrack-CMV-Dsg2-BioID2 plasmid into pAdEasier-1 E. coli cells (Addgene #16399, deposited by Bert Vogelstein; He et al., 1998). Virus packaging and amplification were performed as described (Luo et al., 2007). Virus particles were purified using Vivapure AdenoPACK 20 Adenovirus (Ad5) purification and concentration kit (Sartorius). Adeno-X qPCR Titration Kit (Clontech) was used to calculate virus titer using quantitative PCR on an Applied Biosystems 7900HT.

## Adenovirus infection and biotin labeling

Each experimental replicate included four 35 mm dishes with $10^6$ cells each ($4\times10^6$ total). Cardiomyocytes were infected 1 day after plating with Dsg2–BioID2 adenovirus at a multiplicity of infection (MOI) of 2–5. We reproducibly infected >90% of cardiomyocytes at a low MOI. After 24 h (48 h post plating), the medium was replaced with fresh maintenance medium plus 50 µM biotin in both Dsg2–BioID2-infected and control uninfected samples. The next day (72 h post plating), cells were harvested for protein isolation and MS. Cell lysate preparation and affinity purification were performed according to published protocols (Kim et al., 2016; Le Sage et al., 2016).

## Western blotting

Protein samples were separated on an 8% SDS-PAGE gel and transferred onto a PVDF membrane (Bio-Rad). The membrane was blocked in TBST (20 mM Tris-HCl pH 7.5, 150 mM NaCl, 0.1% Tween-20) plus 5% BSA, washed in TBST, incubated with IRDye 680RD Streptavidin (1:1000, LI-COR) in TBST, washed twice in TBST, and washed once in PBS. The membrane was scanned using a LI-COR Odyssey Infrared Imager.

## Mass spectrometry and statistical analysis

Protein samples were processed using the single-pot, solid-phase-enhanced sample-preparation (SP3) on-bead digestion protocol with trypsin (similar to in Flanagan et al., 2024). In brief, samples were adjusted to a final volume of 200 µl in 100 mM Tris-HCl pH 7.4 and then subjected to reduction with 10 mM dithiothreitol (DTT) and alkylation with 20 mM iodoacetamide (IAA). Subsequently, 1000 µl of 100% acetonitrile (ACN) and 30 µl of 20 mg/ml pre-washed SP3 paramagnetic bead mix (an equal mix of hydrophobic beads (GE Healthcare Bio-Sciences) and hydrophilic beads (GE Healthcare Bio-Sciences) were added, followed by a 20-min incubation with vigorous shaking (1400 rpm). The beads were then washed three times with 80% ethanol, all liquid was removed, and the beads were resuspended in 100 µl of 50 mM ammonium bicarbonate containing 1 µg of trypsin and incubated overnight at 37°C with vigorous shaking (1400 rpm). The supernatant, containing the resulting tryptic peptides, was transferred to a new tube. To ensure complete recovery of peptides, 100 µl of 0.1% trifluoroacetic acid (TFA) was added to wash the SP3 beads, and the wash was combined with the previous supernatant. Tryptic peptides were desalted on a vacuum manifold using 50-mg bed reverse-phase C18 solid phase columns (Supelco), speed-vac dried, and resuspended in 12 µl 0.1% formic acid (FA). A pooled instrument control (PIC) sample was prepared by combining 2 µl from each of the 12 samples and used to monitor instrument reproducibility.

Tryptic peptides were analyzed by nLC-MS/MS using a nanoACQUITY (Waters Corporation) online coupled with an Orbitrap Velos Pro hybrid ion trap mass spectrometer (Thermo Fisher Scientific). For each nLC-MS/MS analysis, 1 µl of peptides was injected onto a $C_{18}$ column PicoChip 25 cm column packed with Reprosil C18 3 µm 120 Å chromatography with a 75 µm ID and 15 µm tip (New Objective). Peptides were eluted off to the mass spectrometer with a 66 min linear gradient of 2–35% ACN/0.1% formic acid at a flow rate of 300 nl/min. The full scan MS spectra were collected over mass range m/z 375–1800 in positive ion mode with an FTMS resolution setting of 60,000 at m/z 400 and AGC target 1,000,000 ms. The top 13 most intense ions were sequentially isolated for collision-induced dissociation (CID) tandem mass spectrometry (MS/MS) in the ion trap with ITMS AGC target 5000 ms. Dynamic exclusion (90 s) was enabled to minimize the redundant selection of peptides previously selected for MS/MS fragmentation.

The nLC-MS/MS data were analyzed with MaxQuant software (Cox and Mann, 2008; Tyanova et al., 2016), version 1.6.6.0. Briefly, the proteomic features were quantified by high resolution full MS intensities after retention alignment and the corresponding MS/MS spectra were searched with Andromeda search engine against the Uniprot mouse database (release November 2017, 82,555 entries; UniProt Consortium, 2011). The mass tolerance was set at 20 ppm for the precursor ions and 0.8 Da for the ITMS fragment ions. Andromeda search included specific trypsin enzyme with a maximum of two missed cleavages, and minimum of seven amino acids in length. Fixed modification carbamidomethyl (C), and variable modifications of oxidation (M), acetyl (protein N-term) and deamidation (NQ) were considered. The protein identification threshold was set to 1% false discovery rate (FDR) as described previously (Cox and Mann, 2008).

Proteins that exhibited statistically significant abundance between DSG2–BioID2 and control sets were selected as follows. Proteins with a single peptide identification were excluded from the data analysis, and unpaired two-tailed Student's t-test on log2 transformed protein intensity was used for the statistical inference to select DSG2–BioID2-interacting proteins. A protein was considered a significant candidate if the t-test P-value was <0.001 and the fold change >10 when compared to the control.

As a surrogate for protein abundance, MaxQuant iBAQ values were used for label-free absolute quantification of identified proteins (Schwanhausser et al., 2011). The average iBAQ value for each protein was determined from the six replicates in both DSG2 and control samples. The final iBAQ value was determined by subtracting the control average from the DSG2 average.

## Bioinformatics analysis

CDH2 BioID proximity proteomics results (ProteomeXchange PXD012496) were previously published by our group (Li et al., 2019). The DSP and the combined JUP, PKP2 and DSC2a $Ca^{2+}$-dependent interactomes (ProteomeXchange PXD037933) were from two previous studies (Badu-Nkansah and Lechler, 2020; Fülle et al., 2024). A similar proximity proteomic

Journal of Cell Science

screen to identify DSG1 interactors (ProteomeXchange PXD036646) in differentiated keratinocytes (Hegazy et al., 2022) was excluded because it focused on trafficking proteins. Venn diagrams comparing protein lists were generated using BioVenn (Hulsen et al., 2008). Canonical signaling pathway enrichment analysis was performed using IPA tools (Qiagen).

## Protein network analysis

The protein interaction map was generated using IPA (Qiagen). Only protein–protein interactions supported by published, experimental data in the manually curated Ingenuity Knowledge Base were considered for building the network. Hierarchical classification was done by grouping the proteins manually using DSG2 at the core. Proteins that bound DSG2 directly were designated as primary interactors. Proteins that bound to primary interactors but not DSG2 were classified as secondary interactors. Proteins that bound secondary interactors were designated as tertiary interactors. Finally, proteins that bound tertiary interactors or to outermost tier proteins were defined as quaternary interactors; 39 proteins could not be linked to the protein network.

## Acknowledgements

We thank Nathan Yates for his assistance with the mass spectrometry experiments.

## Competing interests

The authors declare no competing or financial interests.

## Author contributions

Conceptualization: A.V.K.; Data curation: Y.L., S.B., X.Z., P.S.C., M.S., A.V.K.; Formal analysis: Y.L., A.P.C., S.B., X.Z., E.P., P.S.C., M.S., A.V.K.; Funding acquisition: A.V.K.; Investigation: Y.L., A.P.C., S.B., X.Z., E.P., P.S.C., M.S., A.L.M., A.V.K.; Methodology: Y.L.; Project administration: A.V.K.; Resources: A.V.K.; Supervision: A.V.K.; Writing – original draft: A.V.K.; Writing – review & editing: Y.L., A.L.M., A.V.K.

## Funding

This work was supported by National Institutes of Health R01 HL127711 to A.V.K. and R01 AR072697 to A.L.M. An American Heart Association Predoctoral Fellowship supported S.B. Open Access funding provided by University of Pittsburgh. Deposited in PMC for immediate release.

## Data and resource availability

The mass spectrometry proteomics data have been deposited to the ProteomeXchange Consortium with the dataset identifier PXD073308. All other relevant data and details of resources can be found within the article and its supplementary information.

## Peer review history

The peer review history is available online at https://journals.biologists.com/jcs/lookup/doi/10.1242/jcs.264213.reviewer-comments.pdf

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
