## [Peer Review File · Journal of Cell Science]

The desmoglein 2 interactome in primary neonatal cardiomyocytes

Yang Li, Alexandra P. Campbell, Sahana Balasubramanian, Xuemei Zeng, Emma Porter, Pamela S. Cantrell, Mai Sun, Alexa L. Mattheyses and Adam V. Kwiatkowski
DOI: 10.1242/jcs.264213

Editor: Kathleen Green

Review timeline

Original submission:	12 June 2025
Editorial decision:	16 July 2025
First revision received:	23 November 2025
Accepted:	24 December 2025

Original submission

First decision letter

MS ID#: jcs.264213

MS TITLE: The Desmoglein 2 interactome in primary neonatal cardiomyocytes

AUTHORS: Yang Li; Alexandra Campbell; Sahana Balasubramanian; Xuemei Zeng; Emma Porter; Pamela S Cantrell; Mai Sun; Alexandra Mattheyses; Adam V Kwiatkowski

ARTICLE TYPE: Tools and Resources

Dear Adam,

We have now reached a decision on the above manuscript.

To see the reviewers' reports and a copy of this decision letter, please go to:

As you will see, the reviewers raise a number of substantial criticisms that prevent me from accepting the paper at this stage. They suggest, however, that a revised version might prove acceptable, if you can address their concerns. If you think that you can deal satisfactorily with the criticisms on revision, I would be pleased to see a revised manuscript. We would then return it to the reviewers.

Reviewer 1

Advance summary and potential significance to field

In this manuscript by Li et al, the Kwiatkowski research group identifies the proteins interacting with desmoglein 2 in neonatal cardiomyocytes. The paper largely follows the format of a prior 2019 JSC article exploring the N-cadherin interactome in cardiomyocytes, with this new manuscript doing similar work but in the context of characterizing the desmosome interactome. The prior work has been well cited, suggesting that this work is important and impactful to the field. The impact of this new publication to the field is that the data from the BioID experiments will be useful to other researchers who are studying desmosomes in the heart. The comparison/contrast with adherens junctions also helps to understand the similarities and differences between these two types of cell adhesion complexes.

Comments for the author

Major comments [Please request additional experiments only if they are essential for supporting the conclusions; authors should be encouraged to highlight any claims that are preliminary or speculative, or to discuss any pitfalls or alternative interpretations in a 'Limitations' section]

1. Were the cadherin-2 (N-cad) BioID studies repeated in this work, or are the authors using the previously published 2019 JCS data? The methods doesn't clarify what happened. If it is new data, that should be stated & maybe some analysis to confirm it was similar to prior publications. If it is the old data, can the authors address if the mass-spec approaches are similar enough to enable direct comparisons?
2. The PKP2 tension data (Figure 7) is interesting and compelling. However the authors could have made this data stronger if they had used bioID to show if PKP2 disappears from N-cad vs DSG-2. The idea would be to run N-cad and DSG2 bioID +/- blebbistatin, and then blot for PKP2 to see if blebbistatin has a stronger effect on PKP2 in the adherens junction. As of now, the idea that PKP2 is lost from adherens junctions but retained at desmosomal junctions is speculative. Also the authors did not include any images of cell-cell adhesions stained for adherens junction markers +/- blebbistatin. These are needed to see if the adherens junction is or is not affected by blebbistatin treatment.
3. The FRAP data in Figure 2 is interesting, but requires some statistical testing if the authors wish to make comparisons and state similarities vs differences in this data.
4. The authors speculate that the DSC2b recovery is faster because it has less interactions with cytoskeletal connecting proteins. I was not clear why the authors did not also test DSC2a in their system to confirm this idea? Presumably the DSC2a would recover more slowly. Since some (albeit less) DSC2a is expressed in cardiomyocytes, it would be relevant to understand if this isoform is more stable at cell-cell adhesions.

Minor comments

1. How much further is the biotin ligase from the plasma membrane on DSG2 vs N-cad? Did the authors consider if this distance could affect the interactome of each protein with proteins at the cell-cell junction?

Reviewer 2*Advance summary and potential significance to field*

The authors Li et al., analyzed the desmoglein-2 (DSG2) interactome unique to neonatal cardiomyocytes. Desmosomes and adherens junctions (AJs) are crucial adhesive structures within the intercalated disc (ICD) of cardiomyocytes which is essential for proper mechanical coupling between cardiomyocytes and therefore heart function. This study provides insides in the network responsible for adhesion and signaling at the ICD.

Firstly, the authors identify splice variant DSC2b as the most predominant isoform in cardiomyocytes and analyze the stability/mobility of a set of desmosomal molecules. Secondly, using BioID proximity labeling and quantitative mass spectrometry analysis this study outlines the DSG2 interactome. The authors compare this with published interactomes of junctional components and identify plakoglobin (JUP) and plakophilin (PKP2) as the most abundant shared proteins between CDH/DSG2 interactomes in cardiomyocytes. The authors further conclude that PKP2 is a dynamic protein at cell-cell junctions and whose recruitment of the plasma membrane is tension dependent.

This study provides important insights into the composition of cardiomyocyte junction composition and desmosomal structure in general and can be of interest for the cell biology as well as for the cardiovascular field. However, we find the following points to be addressed to reveal the full potential of the study:

- 1) Even though the authors find DSG2 more stable than other ICD molecules (acc. to Fig 1: 20% mobility in 10min), 24 hours biotin incubation for the bioID screen makes it challenging to dissect interactions at the junction from interactions during DSG2 shuttling and turn-over. How is this affecting the results and interpretation?

- 2) What we can see from the methods, uninfected cells treated with biotin were applied as control for the BioID. How do the authors account for unspecific reactivity of BioID2 or general changes by adenovirus treatment?
- 3) The authors apply neonatal cardiomyocytes to study the interactions of ICD components. The ICD is a complex structure reaching a mature state just after several weeks post-partum in the mouse heart. The authors analyzed the ultrastructure of the ICD in 2D neonatal CM, however, it would be relevant to validate the localization and interaction of at least the top hits in matured heart tissue with biomolecular techniques. Moreover, it's relevant to have some insight on the functionality of the neonatal ICDs (e.g. with respect to mechanical stability, electrical conduction) to allow more general conclusions.
- 4) Further, the study would gain impact if the identified interactome would be further verified by a deeper analysis of the proteins involved in the respective pathways. The authors for example identify pathways unique in certain data subsets and others which are overlaying, like the RhoGTPase signaling pathways. However, they don't comment whether the same proteins within this pathway have been identified. The authors suggest PKP2 as important for junction tension, but a more direct tension measurement, (e.g. via FRET based tension sensors) would be beneficial to strengthen this point. Is Pkp2 here competing with vinculin?
- 5) Several interactome datasets of proteins from different junctional complexes are compared. This would benefit from a graphic representation of the assumed localization/ compartment/ organelle of the respective top hits and their potential interaction.
- 6) The authors give all the necessary technical details to understand the experiments conducted. Although the authors state that the sequences for the plasmids have been verified, they have not been included in the manuscript. Statistical analysis should include checks for normality distribution as the authors use parametric statistics, like ANOVA and t-tests, e.g. for their image analysis. Moreover, an indication of the number of biological and technical replicates and definition of these terms should be provided. In Fig. 1C, min. 3 replicated would be desirable.

Minor:

- page 6, line 40: JUP was analyzed in the previous publication, this does not become clear from the phrasing in the sentence. Better: "taken together with previously published..."
- page 6, line 33: why were cells allowed to grow to semi-confluency for 48-72 h? Was there variation of growth time b/w experiments or constructs?
- page 7, line 19: the authors state that the differences between neonatal CM and MDCK cells solely arise from different inherent desmosomal composition. However, cells are being compared of different organ origin and species. Additionally, here primary cells and a cell line are compared which display a different level of differentiation/specialization.
- Page 8, line 46: please include a western blot for DSG2 and CDH2 not just for the respective protein tag
- Please comment on the intracellular background/cytosolic staining in the IF for streptavidin (Figure 2B)
- Were equal amounts of protein loaded to the SDS-PAGE to check biotinylation? (Figure 2C)
- Page 9, line 13: The authors state that PKP2 is not typically associated with AJ, which is not entirely true as the authors outline in their discussion later. Rephrasing suggested: "...in cardiomyocytes..."
- Page 9, line 34f: The authors state that DSG2 likely recruits a distinct collection of adhesion molecules from CDH2. It is not clear what the authors mean exactly.
- Page 9, line 49: The authors compare two interactomes with each other, but introduce the new abbreviation DM interactome for the ones derived from MDCK cells. Please introduce the abbreviation properly.
- Page 10, line 22: The authors comment on the importance of desmosome in Sertoli cell-germ cell adhesion. Which hits is this comment referring to and have these hits a function in cardiomyocytes or only in the testis?
- Page 10, line 43: Rho GTPase signaling pathway was the only pathway enriched in all four sets. Does this also mean that identical proteins were identified or are different small RhoGTPase pathways altered between the different data sets? (e.g. Cdc42 instead of RhoA...?)
- Page 11, line 4: Please elaborate why 4 tiers were used in the manual construction of this hierarchy. Is there another possible link between the 39 proteins which do not directly or indirectly bind to DSG2 and the constructed network? Maybe signaling-wise?
- Page 11, line 14-16: The authors speculate that the DSG2-only hits dominating the secondary tier include interactors recruited to the desmosome core complex for adhesion and

initiation of signaling. Can you please specify this assumption with linking identified proteins to specific functions? VDAC1/2 for example is a mitochondrial outer membrane marker and PSMD1 is linked to proteasome function, while Cdc42 and DVL3 are polarity markers.

- Page 11, line 34f: The neonatal cardiomyocytes were cultured for different time periods again. Please explain the rationale behind this approach (see Figure 7). Did the authors also perform immunofluorescence stainings for desmosomal markers to distinguish the desmosomal and AJ pool of PKP2?

- Page 11, line 59ff: Please explain the concentration of blebbistatin, which is quite high. Has the concentration been titrated prior the experiment?

- Page 11, line 59ff: Blebbistatin treatment for junctional tension is not a direct readout. Have the authors considered traction force microscopy or FRET based tension sensors?

- Page 28, line 18: the authors write they isolated cDNA from cultured neonatal cardiomyocytes and adult mouse hearts. Elsewhere they state they isolate RNA and then performed RT-PCR to transcribe to cDNA.

First revision

Author response to reviewers' comments

We thank the Editor and the Reviewers for taking the time to review our manuscript. We are particularly grateful to both reviewers for their thoughtful comments and helpful suggestions. We have done our best to address their concerns.

Please find our responses to individual points below. Reviewers' comments are provided *verbatim* in *italics*, and our responses are in **bold**.

Thank you for considering our revised submission.

Reviewer 1

In this manuscript by Li et al, the Kwiatkowski research group identifies the proteins interacting with desmoglein 2 in neonatal cardiomyocytes. The paper largely follows the format of a prior 2019 JSC article exploring the N-cadherin interactome in cardiomyocytes, with this new manuscript doing similar work but in the context of characterizing the desmosome interactome. The prior work has been well cited, suggesting that this work is important and impactful to the field. The impact of this new publication to the field is that the data from the BioID experiments will be useful to other researchers who are studying desmosomes in the heart. The comparison/contrast with adherens junctions also helps to understand the similarities and differences between these two types of cell adhesion complexes.

Major comments

1. Were the cadherin-2 (N-cad) BioID studies repeated in this work, or are the authors using the previously published 2019 JCS data? The methods doesn't clarify what happened. If it is new data, that should be stated & maybe some analysis to confirm it was similar to prior publications. If it is the old data, can the authors address if the mass-spec approaches are similar enough to enable direct comparisons?

The N-cadherin interactome data used for analysis in this study were previously published (Li, Merkel et al., 2019). Identical mass spec approaches were used. We have clarified this in the text (page 7):

“Note that identical mass spectrometry approaches (sample size, sample processing, and data analysis) were used for both studies, permitting a deep and detailed comparison.”

2. The PKP2 tension data (Figure 7) is interesting and compelling. However the authors could have made this data stronger if they had used bioID to show if PKP2 disappears from N-cad vs DSG-2. The idea would be to run N-cad and DSG2 bioID +/- blebbistatin, and then blot for PKP2 to see if blebbistatin has a stronger effect on PKP2 in the adherens junction. As of now, the idea that PKP2 is lost from adherens junctions but retained at desmosomal junctions is speculative. Also the authors did not include any images of cell-cell adhesions stained for adherens junction markers +/- blebbistatin. These are needed to see if the adherens junction is or is not affected by blebbistatin treatment.

We also find the PKP2 tension data intriguing. We are currently using a combination of biochemistry and cell biology to define the different PKP2 complexes and how they are affected by force. Unfortunately, the blebbistatin treatment experiment proposed by the reviewer is not feasible with BioID2. The biotin incubation period (24 hrs) for BioID2 is too long to compare blebbistatin treatments (typically an hour) between constructs. Longer blebbistatin treatments would harm or kill cells and disrupt cell-cell junctions. These experiments would be possible with TurboID (which labels efficiently within 15-30 minutes), but would take considerable work. We feel such experiments are beyond the scope of this work.

Fig. 8 includes images of Plakoglobin (JUP) staining +/- blebbistatin (panels H-J) to mark cell-cell contacts. Based on the reviewer's suggestion, we reanalyzed the data to measure JUP contact/cytoplasm ratios in the presence or absence of blebbistatin. We observed a small but significant increase in JUP contact/cytoplasm ratio in blebbistatin-treated cells. This data is now included in a new supplemental figure (Fig S2). The fact that inhibiting myosin activity increases JUP but decreases PKP2 recruitment strengthens our argument that tension regulates PKP2 recruitment to cell-cell junctions.

3. The FRAP data in Figure 2 is interesting, but requires some statistical testing if the authors wish to make comparisons and state similarities vs differences in this data.

Our view is that FRAP data is semi-quantitative, particularly when analyzing dynamics in beating cardiomyocytes. Nonetheless, we find it extremely useful to provide a broad view of protein/junction stability and reveal gross differences in protein dynamics. Statistical analyses such as the Extra sum-of-squares F test or Akaike's Information Criterion can be used to determine if the best-fit value for specific parameters (e.g., plateau or $t_{1/2}$) differ significantly between two curves. However, while such tests may reveal differences between two fits, we often find that significant differences are unlikely to be biologically relevant (which is why they are not frequently used to compare FRAP data).

Instead, we find it more useful to compare the 95% confidence interval of the recovery mean overlaid with the recovery curve fit. The FRAP plots have been updated to show all data points, the mean recovery at each time point, the 95% confidence interval of the mean at each time point, and the fit (Fig. 2B). We then show the curve fits plus 95% CI for all desmosome proteins analyzed in this manuscript as well as JUP analyzed in our previous manuscript (Fig. 2D). The comparison demonstrates that DSG2, JUP, and DSP have similar recovery profiles (note the degree of overlap), whereas DSC2a, DSB2b, and PKP2 have unique, distinct recovery profiles. We then compare the dynamics of the AJ core (CDH2, CTNNA1, CTNNA1; data from our previous publication) and find that it is similar to the desmosome core.

4. The authors speculate that the DSC2b recovery is faster because it has less interactions with cytoskeletal connecting proteins. I was not clear why the authors did not also test DSC2a in their system to confirm this idea? Presumably the DSC2a would recover more slowly. Since some (albeit less) DSC2a is expressed in cardiomyocytes, it would be relevant to understand if this isoform is more stable at cell-cell adhesions.

The reviewer makes a good suggestion, and we performed additional FRAP experiments to analyze DSC2a dynamics. The new DSC2a data have been added to a revised Figure 2. DSC2a dynamics do indeed differ from DSC2b, and from DSG2 as well (Fig. 2B, D). The DSC2a mobile fraction (33%) lies between DSG2 and DSC2a, and its recovery half-time is faster than both. The faster recovery is puzzling and unexpected, as it suggests the mobile pool is less stable than

DSC2b, even though it possesses a full-length tail. We have revised the results and discussion based on this new data.

Minor comments

1. *How much further is the biotin ligase from the plasma membrane on DSG2 vs N-cad? Did the authors consider if this distance could affect the interactome of each protein with proteins at the cell-cell junction?*

The DSG2 tail is roughly 320 amino acids longer than the N-cadherin tail. The DSG2 tail, like the N-cadherin tail, is unstructured and could “stretch” roughly 120 nanometers further if fully extended. While this is unlikely, the DSG2 tail can theoretically interact with a wider (or at least deeper) range of proteins at the membrane than N-cadherin. We raise this issue in the Results section when comparing the desmosome protein interactomes (pages 8-9).

Reviewer 2

The authors Li et al., analyzed the desmoglein-2 (DSG2) interactome unique to neonatal cardiomyocytes. Desmosomes and adherens junctions (AJs) are crucial adhesive structures within the intercalated disc (ICD) of cardiomyocytes which is essential for proper mechanical coupling between cardiomyocytes and therefore heart function. This study provides insights into the network responsible for adhesion and signaling at the ICD.

Firstly, the authors identify splice variant DSC2b as the most predominant isoform in cardiomyocytes and analyze the stability/mobility of a set of desmosomal molecules. Secondly, using BioID proximity labeling and quantitative mass spectrometry analysis this study outlines the DSG2 interactome. The authors compare this with published interactomes of junctional components and identify plakoglobin (JUP) and plakophilin (PKP2) as the most abundant shared proteins between CDH/DSG2 interactomes in cardiomyocytes. The authors further conclude that PKP2 is a dynamic protein at cell-cell junctions and whose recruitment of the plasma membrane is tension dependent.

This study provides important insights into the composition of cardiomyocyte junction composition and desmosomal structure in general and can be of interest for the cell biology as well as for the cardiovascular field. However, we find the following points to be addressed to reveal the full potential of the study:

- 1) *Even though the authors find DSG2 more stable than other ICD molecules (acc. to Fig 1: 20% mobility in 10min), 24 hours biotin incubation for the bioID screen makes it challenging to dissect interactions at the junction from interactions during DSG2 shuttling and turn-over. How is this affecting the results and interpretation?*

The reviewer raises an important point and a challenge with all BioID2 labeling (and other proximity labeling approaches that require extended labeling times). The long, overnight biotin incubation period labels proteins associated with DSG2, from the ER to the plasma membrane to the endosome. Fortunately, DSG stability (specifically, the large immobile fraction) at cell-cell contacts favors labeling (biotinylation) at cell-cell junctions (note the strong biotinylation at cell-cell contacts in DSG2-BioID2 expressing cells in Fig. 3B). This preferential labeling is also observed in the mass spec data, where adaptor proteins (adaptor and actin-binding adaptor proteins) comprise 45% of the DSG2 interactome by abundance.

We address this issue in a new “Limitations” section of the Discussion (page 14).

- 2) *What we can see from the methods, uninfected cells treated with biotin were applied as control for the BioID. How do the authors account for unspecific reactivity of BioID2 or general changes by adenovirus treatment?*

Uninfected cardiomyocytes incubated with biotin were used as the negative control, the same negative control used in our previously published CDH2 interactome (Li et al., 2019). We considered using a cytoplasmic BioID2 but were concerned that it would 1) eliminate

abundant proteins, like many of the adaptor proteins, and 2) limit comparison with our previous CDH2 dataset. Importantly, cardiomyocytes have low endogenous biotin ligase activity relative to other cell types (Fig. 3C). This low background, combined with the specificity of labeling (primarily junctional), gives us confidence that our screen was selective. Finally, the notable differences between the DSG2 and CDH2 interactomes indicate that, despite similar localization patterns, the probes label distinct protein pools.

3) The authors apply neonatal cardiomyocytes to study the interactions of ICD components. The ICD is a complex structure reaching a mature state just after several weeks post-partum in the mouse heart. The authors analyzed the ultrastructure of the ICD in 2D neonatal CM, however, it would be relevant to validate the localization and interaction of at least the top hits in matured heart tissue with biomolecular techniques. Moreover, it's relevant to have some insight on the functionality of the neonatal ICDs (e.g. with respect to mechanical stability, electrical conduction) to allow more general conclusions.

Our previous studies have established that neonatal cardiomyocytes form robust cell-cell contacts with many of the hallmarks of the ICD, including myofibrils terminating at load-bearing AJs (Li et al., 2025; Li et al., 2019; Merkel et al., 2019). More recently, we have shown that the desmosome (specifically, desmoplakin) is also under load at these developing ICDs (Dong et al., 2025). Furthermore, cultured cardiomyocytes contract spontaneously and form gap junctions (Fig. 1B) to establish electrochemical connections. While the ICD is not fully mature in these cultured cells, it remains a robust system for manipulating and analyzing cardiomyocyte adhesion, organization, and function. We recognize these limitations and discuss them in the Limitations section of the Discussion.

Note also that many of the top hits are established ICD proteins and have been shown to localize to the ICD in mature heart tissue. We plan to pursue these and other hits in future studies.

Further, the study would gain impact if the identified interactome would be further verified by a deeper analysis of the proteins involved in the respective pathways. The authors for example identify pathways unique in certain data subsets and others which are overlaying, like the RhoGTPase signaling pathways. However, they don't comment whether the same proteins within this pathway have been identified. The authors suggest PKP2 as important for junction tension, but a more direct tension measurement, (e.g. via FRET based tension sensors) would be beneficial to strengthen this point. Is Pkp2 here competing with vinculin?

We have now included a deeper analysis of the hits linked to Rho GTPase signaling in the DSG2 and CDH2 interactomes in a revised Fig. 5F. We agree that the tension-dependent role of PKP2 merits further investigation (in fact, we are currently building a FRET-based PKP2 tension sensor), but we feel this falls beyond the scope of the current study.

5) Several interactome datasets of proteins from different junctional complexes are compared. This would benefit from a graphic representation of the assumed localization/compartment/organelle of the respective top hits and their potential interaction.

We have now included a new network map highlighting direct interactions with the core desmosome proteins DSG2, DSC2, PKP2, JUP, and DSP (Figure S1.) We show that the desmosome core interacts directly with at least 50 peripheral proteins (i.e., non-desmosomal proteins) in the DSG2 interactome, nearly 20% of the mapped network in Fig. 7. The large number of direct associations with individual desmosomal proteins underscores the fundamental role of the desmosome complex in adhesion, scaffolding, and signaling.

6) The authors give all the necessary technical details to understand the experiments conducted. Although the authors state that the sequences for the plasmids have been verified, they have not been included in the manuscript. Statistical analysis should include checks for normality distribution as the authors use parametric statistics, like ANOVA and t-tests, e.g. for their image analysis. Moreover, an indication of the number of biological and technical replicates and definition of these terms should be provided. In Fig. 1C, min. 3 replicated would be desirable.

We have updated all Figures and Figure Legends, as well as the Methods, to include detailed information on the number of technical and biological replicates in each experiment and the statistical test used. In addition, we have updated the plot of the blebbistatin experiments in Figure 8 to a SuperPlot (Lord et al., 2020) and added paired t- test estimation plots in a new Figure S2 to provide greater detail into the raw data and statistical differences.

Minor:

- page 6, line 40: *JUP was analyzed in the previous publication, this does not become clear from the phrasing in the sentence. Better: "taken together with previously published..."*

We rephrased the sentence to state explicitly that the JUP dynamics in CMs were published previously by our group (page 5):

"The dynamics of JUP, which binds directly to classical cadherins as part of the AJ, were published previously by our group".

- page 6, line 33: *why were cells allowed to grow to semi-confluency for 48-72 h? Was there variation of growth time b/w experiments or constructs?*

Collecting FRAP data from primary cardiomyocytes is laborious, often requiring 4-6 hrs of imaging per dish (biological replicate). To maximize data acquisition, we typically perform FRAP experiments at 48 hrs and 72 hrs for each construct and pool the data. We only FRAP mature-looking junctions, which are present at both stages, and thus have never observed significant differences between the time points. Note that cardiomyocytes are post-mitotic and do not divide in culture (we also add a mitotic inhibitor to the media to prevent any "contaminating" fibroblasts from dividing and overrunning the culture). They also don't migrate. Therefore, cells are plated at semi-confluency to permit cell-cell attachment with neighboring cells. To clarify, we have reworded the sentence to read (page 5):

"Protein dynamics were measured by fluorescence recovery after photobleaching (FRAP) cells plated at semi-confluency and cultured for 48-72 hours."

- page 7, line 19: *the authors state that the differences between neonatal CM and MDCK cells solely arise from different inherent desmosomal composition. However, cells are being compared of different organ origin and species. Additionally, here primary cells and a cell line are compared which display a different level of differentiation/specialization.*

We have deleted this statement in the revised manuscript.

- Page 8, line 46: *please include a western blot for DSG2 and CDH2 not just for the respective protein tag*

The western blots in Fig. 3C are of streptavidin-precipitated protein isolated from biotin-treated control (uninfected), CDH2-BioID2, or DSG2-BioID2 expressing cardiomyocyte lysates. The blots have two purposes.

1. Using the HA epitope present in BioID2 fusion constructs, we show that DSG2-BioID2 is expressed and full-length in adenoviral infected cells, similar to our previously published CDH2-BioID2 construct (HA blot).
2. We show that the expressed DSG2-BioID2 biotin ligase is active and biotinylates a large collection of proteins in cardiomyocytes, again similar to CDH2-BioID2.

In this context, there was no need to blot for endogenous CDH2 or DSG2.

- *Please comment on the intracellular background/cytosolic staining in the IF for streptavidin (Figure 2B)*

Cytosolic staining of biotinylated proteins is expected. While DSG2-BioID labels proteins at cell-cell contacts (with caveats; see the response to this reviewer's first question about labeling during trafficking above), many of these junctional proteins, especially cytosolic junctional

proteins, have significant “non-junctional” pools. These non-junctional pools could be other cellular regions/compartments (e.g., focal adhesions, z-discs) or simply cytoplasmic. For an example of protein shuttling between the cell-cell contacts and the z-disc, see our CDH2 interactome paper (Li et al., 2019).

- Were equal amounts of protein loaded to the SDS-PAGE to check biotinylation? (Figure 2C)

Equal amounts of protein lysate were used for the initial streptavidin pulldowns, but the amount of blotted protein varied depending on the amount of biotinylated protein (i.e., given the low level of endogenous biotinylation in the control cells, less protein was precipitated and blotted compared the CDH2 and DSG2 samples). As noted above, the point of this panel was to demonstrate DSG2-BioID functionality (expression and biotin ligase activity) in cardiomyocytes, not to directly compare expression/labeling levels.

- Page 9, line 13: The authors state that PKP2 is not typically associated with AJ, which is not entirely true as the authors outline in their discussion later. Rephrasing suggested: “...in cardiomyocytes...”

We reworded the sentence to read (page 8):

“PKP2, however, is a core component of the desmosome and does not bind directly to classical cadherins.”

- Page 9, line 34f: The authors state that DSG2 likely recruits a distinct collection of adhesion molecules from CDH2. It is not clear what the authors mean exactly.

We have deleted this sentence and added a new paragraph (and new panel F to Figure 5) that delves deeper into the Rho GTPase signaling pathways associated with DSG2 and/or CDH2 hits (page 8).

- Page 9, line 49: The authors compare two interactomes with each other, but introduce the new abbreviation DM interactome for the ones derived from MDCK cells. Please introduce the abbreviation properly.

We have clarified the abbreviation and changed it to “Desmo” in the text and figure.

- Page 10, line 22: The authors comment on the importance of desmosome in Sertoli cell-germ cell adhesion. Which hits is this comment referring to and have these hits a function in cardiomyocytes or only in the testis?

The point here was to note that Sertoli cell-germ cell interactions were enriched in the DSG2-Desmo shared hits. We have revised the sentence to read:

“SG-GC junction signaling (as well as GC-SC junction signaling) pathways were enriched in all sets, reflecting the importance of desmosome proteins in the unique, but poorly understood, desmosome-like junction that connects Sertoli cells to germ cells in the testes (Lie et al., 2011).”

- Page 10, line 43: Rho GTPase signaling pathway was the only pathway enriched in all four sets. Does this also mean that identical proteins were identified or are different small RhoGTPase pathways altered between the different data sets? (e.g. Cdc42 instead of RhoA...?)

Good question. Slightly different Rho signaling pathways are enriched in the different desmosome data sets. Notably, the DSG2 unique hits were enriched for proteins that directly regulate Rho GTPase activity and included the Rho GTPase CDC42. We have revised this section to read (pages 9-10):

“Rho GTPase cycle signaling was also enriched in the DSG unique hits (Fig. 6B), and this was the only pathway observed in all four sets. Notably, the DSG unique hits linked to Rho GTPase cycle included the Rho GTPase CDC42 and proteins that regulate Rho GTPase activity, such as

ARHGAP1 (Rho GTPase Activating Protein 1), ARHGDI1 (Rho GDP Dissociation Inhibitor Alpha), and RASAL2 (RAS Protein Activator Like 2). Rho signaling unique to the DSG2 interactome could reflect a cardiomyocyte-specific repertoire of Rho GTPase regulatory proteins necessitated by the unique cytoskeletal organization and physical demands of cardiomyocyte cell-cell adhesion.”

- Page 11, line 4: Please elaborate why 4 tiers were used in the manual construction of this hierarchy. Is there another possible link between the 39 proteins which do not directly or indirectly bind to DSG2 and the constructed network? Maybe signaling-wise?

Hierarchical classification was done manually around DSG2. We used a similar organization when mapping the CDH2 interactome (Li et al., 2019). Primary interactors bind DSG2 directly. Secondary interactors bind to primary interactors, but not to DSG2. Tertiary interactors bind secondary interactors, and quaternary interactors bind tertiary interactors. These details are provided in the Figure Legend and Methods.

There are no obvious links between the 39 proteins (Table S2) that were not connected to the larger network.

- Page 11, line 14-16: The authors speculate that the DSG2-only hits dominating the secondary tier include interactors recruited to the desmosome core complex for adhesion and initiation of signaling. Can you please specify this assumption with linking identified proteins to specific functions? VDAC1/2 for example is a mitochondrial outer membrane marker and PSMD1 is linked to proteasome function, while Cdc42 and DVL3 are polarity markers.

A more detailed analysis of these secondary interactions is the focus of current research. Rather than speculate further, we feel it better to provide this initial analysis and allow the reader to use the data to build and test their own models for desmosome signaling and function.

- Page 11, line 34f: The neonatal cardiomyocytes were cultured for different time periods again. Please explain the rationale behind this approach (see Figure 7). Did the authors also perform immunofluorescence stainings for desmosomal markers to distinguish the desmosomal and AJ pool of PKP2?

Cardiomyocytes were fixed at 72-96 hours post-plating to enrich for mature junctions, versus 48-72 hrs when examining protein dynamics by FRAP. As noted above, junction maturity varies between cardiomyocytes within a given prep/dish. When conducting live- cell FRAP experiments, we selectively choose mature-looking junctions between actively contracting cardiomyocytes expressing the transfected protein of interest for analysis. In fixed cells, we cannot screen this way; therefore, we “enrich” for mature junctions by fixing cells at a later timepoints (72 and 96 hrs) and examining endogenous protein by immunolabeling.

When performing the colocalization analyses in Figure 8, we lacked a robust antibody to a desmosome-specific protein such as DSG2 or DSP. Nonetheless, the point of Figure 8 was to determine if PKP2 colocalizes with AJ proteins like CTNNA1 and CTNNB1, which it does.

- Page 11, line 59ff: Please explain the concentration of blebbistatin, which is quite high. Has the concentration been titrated prior the experiment?

Blebbistatin is typically used in the range of 10 to 100 μM for cell culture experiments. The concentration we use (100 μM) is at the high end of this range, but it's a concentration we reached after testing a range of concentrations (1 to 100 μM) for effective inhibition of myosin activity in cardiomyocytes. 100 μM works best in our system, perhaps due to the high levels of myosin activity, and we have observed no cytotoxic effects if used for one hour or less ((Dong et al., 2025; Merkel et al., 2019).

- Page 11, line 59ff: Blebbistatin treatment for junctional tension is not a direct readout. Have the authors considered traction force microscopy or FRET based tension sensors?

While we are currently testing various FRET-based tension sensors in our cells, we have recently published a paper that shows blebbistatin treatment relieves force-induced opening of desmoplakin in cardiomyocytes (Dong et al., 2025). We also showed previously that blebbistatin reduces vinculin and afadin recruitment to cardiomyocyte AJs (Merkel et al., 2019).

- Page 28, line 18: the authors write they isolated cDNA from cultured neonatal cardiomyocytes and adult mouse hearts. Elsewhere they state they isolate RNA and the performed RT-PCR to transcript to cDNA.

Thank you for catching this mistake. We isolated RNA from cardiomyocyte and use RT-PCR to make cDNA. We have corrected the text.

References

- Dong, Y., Elgerbi, A., Xie, B., Han, Y., Kwiatkowski, A. V., Choy, J. S. and Sivasankar, S. (2025). Actomyosin forces trigger a conformational change in desmoplakin within desmosomes. *Nat Commun* **16**, 9052.
- Li, X., Wortelboer, R., Song, Y., Balasubramanian, S., McLain, C., Manriquez, A. H., Suh, J. D., Hoffman, B. D., Kwiatkowski, A. V. and Radice, G. L. (2025). Vinculin Y822 phosphorylation regulates adhesion remodeling during cardiomyocyte maturation. *J Cell Sci*.
- Li, Y., Merkel, C. D., Zeng, X., Heier, J. A., Cantrell, P. S., Sun, M., Stolz, D. B., Watkins, S. C., Yates, N. A. and Kwiatkowski, A. V. (2019). The N-cadherin interactome in primary cardiomyocytes as defined using quantitative proximity proteomics. *J Cell Sci* **132**.
- Lie, P. P., Cheng, C. Y. and Mruk, D. D. (2011). The biology of the desmosome- like junction a versatile anchoring junction and signal transducer in the seminiferous epithelium. *Int Rev Cell Mol Biol* **286**, 223-69.
- Lord, S. J., Velle, K. B., Mullins, R. D. and Fritz-Laylin, L. K. (2020). SuperPlots: Communicating reproducibility and variability in cell biology. *J Cell Biol* **219**.
- Merkel, C. D., Li, Y., Raza, Q., Stolz, D. B. and Kwiatkowski, A. V. (2019). Vinculin anchors contractile actin to the cardiomyocyte adherens junction. *Mol Biol Cell* **30**, 2639-2650.

Second decision letter

MS ID#: jcs.264213R1

MS Title: The Desmoglein 2 interactome in primary neonatal cardiomyocytes

Authors: Yang Li; Alexandra Campbell; Sahana Balasubramanian; Xuemei Zeng; Emma Porter; Pamela S Cantrell; Mai Sun; Alexa Mattheyses; Adam V Kwiatkowski

Article Type: Tools and Resources

Dear Adam,

I am happy to tell you that your manuscript has been accepted for publication in Journal of Cell Science, pending standard publication integrity checks.